# SAE as a Crystal Ball 🔮:
# Interpretable Features Predict Cross-domain Transferability of LLMs without Training

**Qi Zhang**[1,2][*] **Yifei Wang**[3][*] **Xiaohan Wang**[2] **Jiajun Chai**[2]
**Guojun Yin**[2] **Wei Lin**[2] **Yisen Wang**[1,4][†]

[1] State Key Lab of General Artificial Intelligence,
  School of Intelligence Science and Technology, Peking University
[2] Meituan [‡]
[3] Amazon AGI SF Lab[§]
[4] Institute for Artificial Intelligence, Peking University

## Abstract

In recent years, pre-trained large language models have achieved remarkable success across diverse tasks. Besides the pivotal role of self-supervised pre-training, their effectiveness in downstream applications also depends critically on the post-training process, which adapts models to task-specific data and objectives. However, this process inevitably introduces model shifts that can influence performance in different domains, and how such shifts transfer remains poorly understood. To open up the black box, we propose the SAE-based Transferability Score (STS), a new metric that leverages sparse autoencoders (SAEs) to forecast post-training transferability. Taking supervised fine-tuning as an example, STS identifies shifted dimensions in SAE representations and calculates their correlations with downstream domains, enabling reliable estimation of transferability *before* fine-tuning. Extensive experiments across multiple models and domains show that STS accurately predicts the transferability of supervised fine-tuning, achieving Pearson correlation coefficients above 0.7 with actual performance changes. Beyond this, we take an initial step toward extending STS to reinforcement learning. We believe that STS can serve as an interpretable tool for guiding post-training strategies in LLMs. Code is available at `https://github.com/PKU-ML/STS`.

## 1 Introduction

Recent advances in large-scale neural networks have demonstrated that pre-training on massive datasets yields models with strong generalization capabilities (Achiam et al., 2023; Grattafiori et al., 2024; Yang et al., 2025; Liu et al., 2024). However, due to discrepancies between the pretraining objectives and the specific requirements of downstream tasks, pretraining alone is often insufficient to achieve optimal performance on these tasks. Post-training, which includes supervised fine-tuning (Zhang et al., 2023; Luo et al., 2023), and reinforcement learning (Schulman et al., 2017; Shao et al., 2024; Li et al., 2025b), play a critical role in bridging this gap. By selectively adapting the pretrained model, post-training improves performance in target tasks, and allows models to better capture domain-specific characteristics.

However, during the post-training process, it is widely observed that improvements on a target task often come at the expense of performance in other domains (Dong et al., 2023; Kumar et al., 2022). For instance, Li et al. (2025a) state that improvements in the reasoning ability of large language

---

[*]Equal Contribution.

[†]Corresponding Author: Yisen Wang (yisen.wang@pku.edu.cn).

[‡]Qi Zhang completed this work during an internship at Meituan.

[§]This work was completed at MIT prior to Yifei Wang joining Amazon.

models come at the cost of reduced model robustness. Despite these observations, the mechanisms underlying how model features are linked and transferred during post-training remain largely unexplored. As a result, we currently lack the ability to predict which domain performance is likely to benefit or deteriorate under specific post-training adaptations, limiting both interpretability and principled design of post-training strategies.

In this paper, we analyze the transferability of post-training through the enhanced interpretability provided by sparse autoencoders (Ng et al., 2011). The sparse autoencoder (SAE) is an encoder-decoder architecture that reconstructs the internal activations of models while enforcing sparsity constraints on the hidden layer. Previous works have shown that the SAE encoder features achieve monosemanticity (Cunningham et al., 2023; Gao et al., 2024; Zhang et al., 2025), where each dimension is only activated by a certain natural concept. Leveraging this property, we observe that post-training only modifies certain SAE dimensions—for example, those associated with mathematical reasoning. This observation motivates a natural approach to predict the transferability of post-training: we can identify the shifted SAE features and examine their correlations with different domains.

Concretely, our analysis consists of two steps: (1) identifying the dimensions that are shifted during post-training, and (2) assessing their correlations with downstream domains. In the first stage, the primary challenge is to identify the shifted dimensions **prior to post-training**. Inspired by the observation that in-context learning exhibits behaviors similar to supervised fine-tuning (Wang et al., 2023; Mosbach et al., 2023), we forecast the shifted dimensions by using the supervised answers as demonstrations for in-context learning, and then identify the dimensions that undergo the largest changes. Empirical results show a clear overlap between the predicted and actual shifted dimensions. In the second stage, leveraging the interpretability of SAE activations, we observe that the activation values of these shifted dimensions in a domain can capture their correlation. We formalize this as the SAE-based transferability score (STS), which quantifies how strongly the shifted dimensions relate to downstream tasks. A higher STS suggests a larger expected performance change after supervised fine-tuning. Empirically, we find that our metric consistently correlates well with actual performance shifts. For instance, the Pearson correlation coefficient exceeds 0.7 when evaluating performance variations across domains in the MMLU-Pro dataset (Wang et al., 2024). At last, we provide a preliminary exploration of extending our metric to reinforcement learning settings. Together, these results allow us to develop an interpretable framework for predicting the cross-domain transferability without training. We summarize our contributions as follows:

- We propose a method to identify shifted dimensions in supervised fine-tuning without requiring access to the fine-tuned models. We observe that when supervised answers are used as context prompts, the shifted dimensions in in-context learning substantially overlap with those in supervised fine-tuning.

- We propose the SAE-based Transferability Score (STS), which uses correlations in SAE feature space and estimated shifted dimensions to accurately predict LLM transferability without performing supervised fine-tuning.

- We empirically show that higher STS values strongly correlate with larger performance shifts in supervised fine-tuning, achieving Pearson correlations above 0.7 across diverse scenarios. This confirms that STS is a reliable, fine-tuning-free metric for predicting LLM cross-domain transferability.

## 2 RELATED WORK & PRELIMINARY

**Post-training.** Post-training refers to the stage after large-scale pretraining, where a pretrained model is further adapted to align with specific objectives, user preferences, or downstream applications (Ouyang et al., 2022; Rafailov et al., 2023; Yu et al., 2025). The methods in post-training can be majorly divided into supervised fine-tuning (SFT) and reinforcement learning (RL). During the SFT process, given a set of labeled examples $\{x_i, y_i\}$, the model parameters are updated to minimize the discrepancy between the model's predictions and the ground-truth answers via Negative Log-Likelihood (NLL) Loss:

$$\mathcal{L}_{\text{SFT}}(\Theta) = -\mathbb{E}_{x_i} \log p(y_i | x_i; \Theta).$$

where $\Theta$ denotes the model parameters. After SFT, RL is used to further align a pretrained model with human preferences or task-specific objectives. Concretely, the model policy $p(y|x; \Theta)$ is optimized to maximize a reward signal from a reward model or rule-based function:

$$J(\Theta) = \mathbb{E}_{y \sim p(y|x;\Theta)}[r(x, y)].$$

In post-training research, transferability has long been a central focus. For example, Huan et al. (2025) empirically examines the transfer of capabilities from mathematical reasoning to other downstream tasks, while Sun et al. (2025) offers a more fine-grained assessment of out-of-distribution generalization performance. Additionally, Chu et al. (2025) compares the generalization behaviors of SFT and RL-based post-training methods. However, most of the current works focus on post-hoc analysis after training. As a result, such approaches are less practical because they cannot help predict transfer effects before the fine-tuning process starts. This limitation motivates our work, where we aim to build a method that can predict transferability without fine-tuning.

**Sparse Autoencoders.** Although large language models (LLMs) have demonstrated remarkable performance across a wide range of downstream tasks, many of their decisions and internal behaviors remain opaque, which hinders broader deployment in applications. To address this issue, sparse autoencoders (SAEs) have been proposed as a promising framework for improving the mechanistic interpretability of LLMs (Lieberum et al., 2024; Cunningham et al., 2023). Concretely, given a hidden representation $z \in \mathbb{R}^d$ within the network, an SAE employs an encoder–decoder architecture to project $z$ into a sparse latent representation and reconstruct it back to the original space. For instance, in the case of a top-K SAE (Gao et al., 2024), the encoding-decoding process can be formulated as:

$$\begin{aligned} h &= \text{TopK}(W_e z - b), \\ \hat{z} &= W_d h + b. \end{aligned} \quad (1)$$

The encoder representation $h$ is computed via a linear transformation defined by $W_e \in \mathbb{R}^{s \times d}$ and a bias $b \in \mathbb{R}^s$, while the decoder reconstructs the input features using $W_d \in \mathbb{R}^{d \times s}$. The SAE is trained by minimizing the reconstruction loss:

$$\mathcal{L}_{\text{SAE}}(W_e, W_d, b) = \|\hat{z} - z\|^2.$$

Previous studies (Gao et al., 2024) have shown that when the encoder features are sufficiently sparse (e.g., $K \ll s$), the resulting representations often display monosemanticity. In other words, each feature dimension is only activated by a certain natural concept, such as a mathematical definition, a physical property, or a linguistic pattern.

**In-context Learning.** In-context learning denotes the ability of large pretrained models to solve tasks by conditioning on demonstrations provided in the input (Kossen et al., 2023; Wang et al., 2025). Formally, given a context consisting of $k$ labeled examples

$$\mathcal{C} = \{(x_1, y_1), (x_2, y_2), \ldots, (x_k, y_k)\},$$

the model receives a new query input $x_{k+1}$ and generates the output $\hat{y}_{k+1}$ by leveraging the conditional distribution learned during pretraining:

$$\hat{y}_{k+1} \sim p_\theta(y \mid x_{k+1}, \mathcal{C}),$$

where $\theta$ denotes the fixed pretrained parameters. Unlike supervised fine-tuning and reinforcement learning, in-context learning (ICL) adapts to new tasks during inference without requiring additional training. Nevertheless, several studies have shown that models under in-context learning still exhibit many similarities to those trained with supervised fine-tuning and reinforcement learning (Mosbach et al., 2023; Wang et al., 2023).

## 3 Capturing Shifted Features in Supervised Fine-tuning

In this paper, we analyze the transferability of supervised fine-tuning across different domains by understanding how feature representations shift within neural networks. Since sparse autoencoders (SAEs) enhance monosemanticity by disentangling overlapping representations, their features provide a clearer interpretability for tracking representation shifts in different domains. Consequently, in this section, we start by analyzing how the supervised fine-tuning process modifies SAE features and how the shifted features can be predicted in advance of the fine-tuning process.

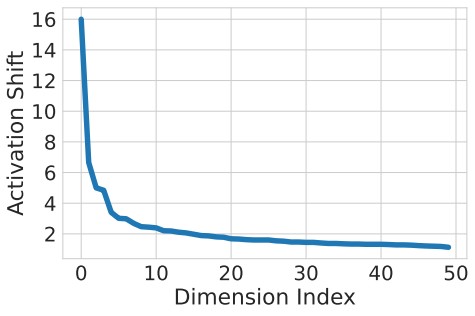

(a) Activation Shifts on SAE Dimensions

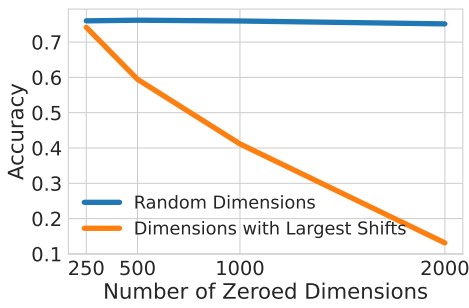

(b) Test Accuracy with Zeroed SAE Dimensions

Figure 1: Analysis of feature shifts induced by supervised fine-tuning (SFT). We fine-tune Qwen2.5-7B-Instruct on the LIMO (a mathematical reasoning dataset) and examine shifts of SAE features on the residual stream at layer 25. Figure (a) shows the distribution of shift magnitudes while Figure (b) shows accuracy on Math-LightEval when progressively zeroing the dimensions with the largest shifts. The results indicate that SFT primarily affects a small subset of SAE dimensions tied to specific model capabilities.

### 3.1 SFT-INDUCED CHANGES IN SAE FEATURES

Prior works have shown that SAE encoder features exhibit strong interpretability, with each dimension corresponding to a certain natural concept. Since supervised fine-tuning (SFT) is generally tailored to specific downstream tasks and targeted capabilities, we wonder whether it primarily affects only a small subset of SAE dimensions tied to task-relevant features. Taking the mathematical ability as an example, we investigate how SAE features change when models are fine-tuned on the math dataset LIMO (Ye et al., 2025). As noted in (Lieberum et al., 2024), the same dimension in an SAE usually continues to represent similar concepts after fine-tuning. Consequently, we extract monosemantic features before and after the SFT process using the same SAE on the residual streams of 25-th layer in Qwen2.5-7B-Instruct (Team, 2024).

As shown in Figure 1a, we find that changes in SAE features are largely concentrated in a small subset of dimensions. With calculation, we find that the top-100 dimensions account for 25% of the total change, indicating that the SFT process primarily affects only a limited portion of SAE features. Furthermore, to observe the relationship between features shifted during SFT and the mathematical ability of the model, we rank SAE features according to their magnitude of change during fine-tuning and then evaluate model performance on the Math-Lighteval dataset (Hendrycks et al., 2021) by zeroing out different numbers of features. As shown in Figure 1b, performance on math tasks drops rapidly when the shifted features are removed, whereas the model retains strong mathematical abilities when random SAE features are zeroed. These results indicate that the SFT process primarily changes a small subset of SAE features that are closely associated with specific model capabilities.

### 3.2 IDENTIFYING SHIFTED DIMENSIONS VIA IN-CONTEXT LEARNING

As previously discussed, supervised fine-tuning modifies only a small subset of SAE features that correspond to specific semantics. Intuitively, these features are crucial for studying properties of SFT, such as transferability. However, their identification typically requires examining the model after fine-tuning, which confines the analysis to a post-hoc perspective. This limitation motivates a central question of our work: can such features be identified prior to the fine-tuning process, thereby enabling a predictive understanding of transferability?

To solve this challenge, we draw on the connection between supervised fine-tuning (SFT) and in-context learning (ICL). Previous works demonstrate that ICL can obtain similar performance to SFT in the large language models (Wang et al., 2023; Mosbach et al., 2023). Consequently, this motivates us to investigate whether the SAE features shifted during ICL and SFT are consistent. To verify this hypothesis, we respectively sort the SAE dimensions according to their changes after SFT and ICL. To be specific, we conduct experiments on Qwen2.5-7B-Instruct. For SFT, we employ ground-truth

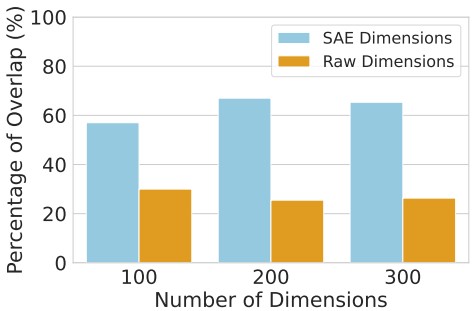 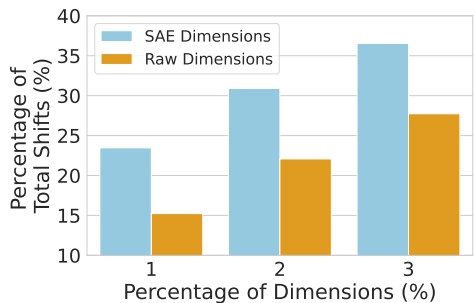

(a) Degrees of Overlaps between Estimated and Actual Shifted Dimensions

(b) Percentage of Overall Activation Change from Top-Shifted Dimensions

Figure 2: Overlap between estimated dimensions and the training task. Figure (a) demonstrates that the SAE shifted dimensions predicted by ICL substantially overlap with the actual shifted dimensions identified after SFT, whereas applying the same method directly on raw dimensions is less effective. Figure (b) further shows that raw model dimensions, prior to applying SAE, are influenced more uniformly by the SFT process, thereby limiting the ability to identify crucial shifted dimensions.

chain-of-thoughts (CoTs) on LIMO as supervision, while for ICL, we use the same CoTs as context prompts. As shown in Figure 2a, we observe substantial overlap between the shifted features in SFT and ICL; for example, 57% of the top 100 most shifted SAE dimensions coincide. These findings confirm that **the shifted dimensions in ICL and SFT are highly consistent**, suggesting that the shifted dimensions can be identified before the SFT process.

We have shown that, by leveraging the connection between ICL and SFT, the shifted SAE dimensions can be identified prior to the SFT process. A natural question then arises: are SAEs necessary for this method? Intuitively, as dimensions in the raw feature space [1] are highly polysemantic and entangled (Elhage et al., 2022), specific abilities of the model (e.g., mathematical reasoning) are distributed across multiple dimensions. As a result, the SFT process tends to affect a broader set of features, which makes it more difficult to identify crucial shifted dimensions. To verify the analysis, we further conduct experiments on the raw features before applying SAEs. We again sort the features by their changes during SFT and ICL. As shown in Figure 2b, the shifted features before SAE are more uniformly distributed. We then investigate the influence on the accuracy of identifying shifted dimensions. With the same empirical settings, Figure 2a shows that the overlap between shifted features in ICL and SFT is much reduced in the raw feature space. These findings indicate that the enhanced monosemanticity introduced by SAEs is crucial for identifying shifted features prior to supervised fine-tuning.

## 4 PREDICTING THE TRANSFERABILITY OF SUPERVISED FINE-TUNING ACROSS DOMAINS

In Section 3, we introduced a method for predicting shifted SAE dimensions prior to the SFT process. Intuitively, when applying fine-tuned models to downstream tasks, if the shifted dimensions are closely related to a given domain, the SFT influence on that domain will be stronger. Thus, understanding the transferability of SFT across domains requires analyzing the correlations between shifted dimensions and different domains. In this section, we present a metric for evaluating this correlation and predict transferability based on that. Specifically, Section 4.1 introduces the proposed metric, while Section 4.2 validates it across diverse scenarios.

### 4.1 METRICS FOR MEASURING CROSS-DOMAIN CORRELATIONS

Due to the enhanced monosemanticity of SAEs (Cunningham et al., 2023), the activations in the SAE feature space become interpretable, meaning that the top-activated sequences within a given

---

[1]The raw model dimensions refer to the representations that are used as the input of the SAEs.

dimension usually share similar semantics. This property allows us to associate each dimension with specific semantic concepts. Consequently, if the sequences from a particular domain exhibit higher activation values in a certain dimension, it implies that this dimension is more strongly correlated with that domain. Building on this intuition, we can use the degrees of activations across dimensions to quantify domain–feature correlations and further analyze the transferability of supervised fine-tuning.

Formally, we define our metric as the SAE-based transferability score (STS). In the first step, given an SFT dataset $\mathcal{T} = \{x_i, y_i\}$, we extract SAE features for each sample. Let the SAE features be denoted as $h(x_i; \Theta)$, where $\Theta$ represents the parameters of the pretrained model, and $h$ is the SAE encoder trained on top of the pretrained model. Similarly, with in-context learning, the features are denoted as $h(x_0, y_0, \cdots, x_t, y_t, x_i; \Theta)$, where $\{x_0, y_0, \cdots, x_t, y_t\}$ are the context prompts. We then identify the $N$ dimensions with the largest changes, i.e.,

$$D_N = \text{TopN}(\mathbb{E}_{x_i} \| h_j(x_i; \Theta) - h_j(x_0, y_0, \cdots, x_t, y_t, x_i; \Theta) \|^2),$$

where $h_j$ denotes the $j$-th dimension of the SAE features.

In the second step, given a downstream domain dataset $\tilde{\mathcal{T}} = \{\tilde{x}_i\}$, we compute the activation values on the shifted dimensions identified in the first step:

$$\text{STS}_{\text{act}}(\tilde{\mathcal{T}}) = \mathbb{E}_{\tilde{x}_i} \sum_{j \in D_N} h_j(\tilde{x}_i; \Theta).$$

It is important to note that **we do not use the model after SFT in this estimation**, which means that our metric serves as a predictive measure rather than a post-hoc analysis.

Besides directly computing the average activation values, we introduce an alternative method based on in-context learning (ICL) to capture the correlation between the downstream domain and the shifted dimensions. We know that ICL leverages multiple demonstrations to guide the model in performing downstream tasks, effectively injecting domain-specific signals into the representation. Intuitively, by comparing the SAE features extracted with and without ICL demonstrations, we can isolate the effect of domain knowledge on the shifted dimensions. This difference reflects how strongly the features are modulated by task-relevant information, thereby offering a reliable estimate of the correlation between a domain and the shifted dimensions.

Consequently, we estimate the correlation by measuring the difference between features extracted with and without in-context demonstrations. Formally, let $\{\tilde{x}_i, \tilde{y}_i\}_{i=1}^m$ denote $m$ ground-truth question-answer pairs in the downstream domain. We define the metric as

$$\text{STS}_{\text{ICL}}(\tilde{\mathcal{T}}) = \mathbb{E}_{\tilde{x}_i} \sum_{j \in D_N} \| h_j(\tilde{x}_0, \tilde{y}_0, \cdots, \tilde{x}_m, \tilde{y}_m, \tilde{x}_i; \Theta) - h_j(\tilde{x}_i; \Theta) \|^2.$$

This metric captures how much the presence of domain-relevant context (the demonstrations) influences the shifted dimensions, providing another reliable estimate of domain-feature correlation besides using maximum activation values.

## 4.2 EMPIRICAL VERIFICATIONS ON PREDICTING THE TRANSFERABILITY

Based on the proposed metric, we now empirically evaluate the correlation between the STS score and the actual transferability across different downstream domains. Specifically, we first use the LIMO dataset (Ye et al., 2025) as the SFT training set, which contains 817 high-quality mathematical examples. We fine-tune three models (Qwen2.5-7B-Instruct, Llama3-8B-Instruct, and Gemma2-9B-Instruct) on LIMO, and compare their performances before and after SFT on different domains of MMLU-Pro (Wang et al., 2024). To extract SAE features, we apply SAEs to the residual streams of the models prior to SFT. For computing the STS metric, we employ two ground-truth CoTs from LIMO as in-context demonstrations to identify the top-100 shifted dimensions. When estimating the correlations between the domains and the predicted shifted dimensions, we use five ground-truth CoTs from the domain of MMLU-Pro as prompts to calculate $\text{STS}_{\text{ICL}}$. More details of the experiments can be found in the Appendix.

As shown in Figure 3, the correlation between STS and performance changes across different domains remains consistently high for all three models, with Pearson correlation coefficients exceeding

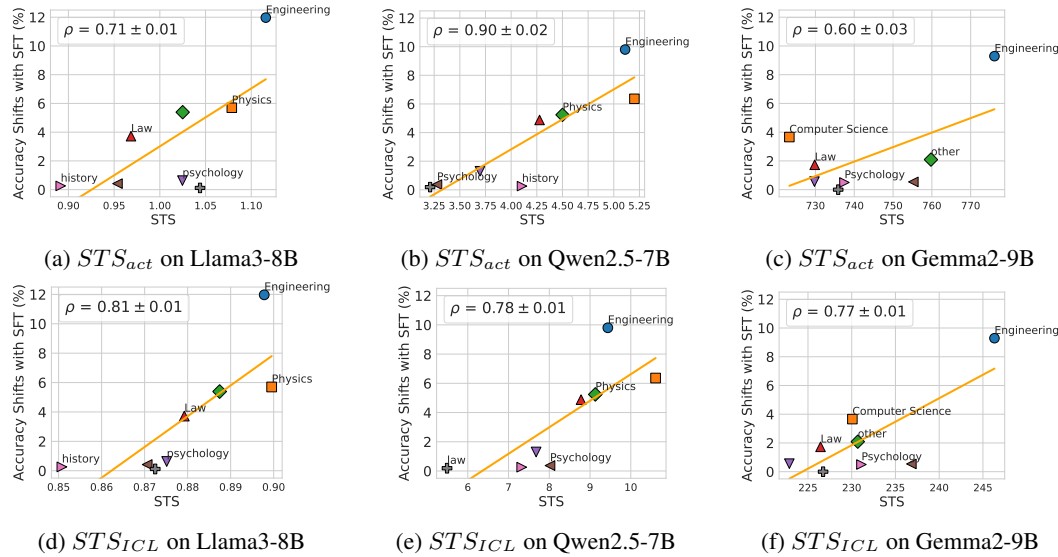

Figure 3: The Pearson correlation ($\rho$) between STS and actual absolute performance shifts on MMLU-Pro induced by SFT on LIMO. Each experiment is repeated three times, and we report the mean and standard deviation of $\rho$; the fitted line shown corresponds to one of the runs. We extract SAE features from Llama3-8B-Instruct, Qwen2.5-7B-Instruct, and Gemma2-9B-Instruct. During the evaluation process, we select four MMLU-Pro domains with the largest and smallest performance shifts under SFT. The detailed performance shifts can be found in Appendix A.

Table 1: Verification on a code dataset (Verifiable-Coding-Problems-Python-10k-Dataset) and a clinical reasoning dataset (CoT_Reasoning_Mens_Mental_Health). The Qwen-2.5-7B-Instruct model is trained on each dataset, and we evaluate: (1) the correlation between ICL feature shifts and SFT, and (2) the correlation between actual performance shifts and our proposed metric.

| Training Domain | Correlations between Actual Performance Shifts and STS$_{\text{ICL}}$ | Overlap between Top 100 Estimated and Actual Shifted Dims |
|---|---|---|
| Code | $0.77 \pm 0.01$ | 62 |
| Health | $0.71 \pm 0.02$ | 57 |

60%. These findings validate that our metric provides a reliable estimation of the transferability of the SFT process. Besides, we note that the performance of STS$_{\text{ICL}}$ is more stable than STS$_{\text{act}}$ (the coefficients keeps above 75%), which suggests that leveraging in-context learning yields a more accurate estimation of the correlation between domains and SAE dimensions than relying solely on activation values.

To better verify the effectiveness of our metric, we further conduct the experiments in different domains to provide more empirical support. Specifically, we include a code-generation dataset (Verifiable-Coding-Problems-Python-10k-Dataset (Hugging Face, 2025)) and a health dialogue dataset (CoT_Reasoning_Mens_Mental_Health (Wesney, 2025)) to examine how well our metric STS correlates with actual downstream performance changes. In addition, we further evaluate the overlap between ICL-induced feature drift and SFT-induced shifts. Taking Qwen2.5-7B-Instruct as an example, the empirical results are summarized in Table 1. As shown in the table, we observe a strong correlation between our proposed metric and actual performance shifts across different datasets. In addition, our central hypothesis that there is a substantial overlap between ICL and SFT shifted dimensions in the SAE representation space **continues to hold across different training datasets**. These results reinforce the validity of our method and expand the scope of our method by demonstrating its effect across diverse training domains.

## 4.3 ABLATION STUDY

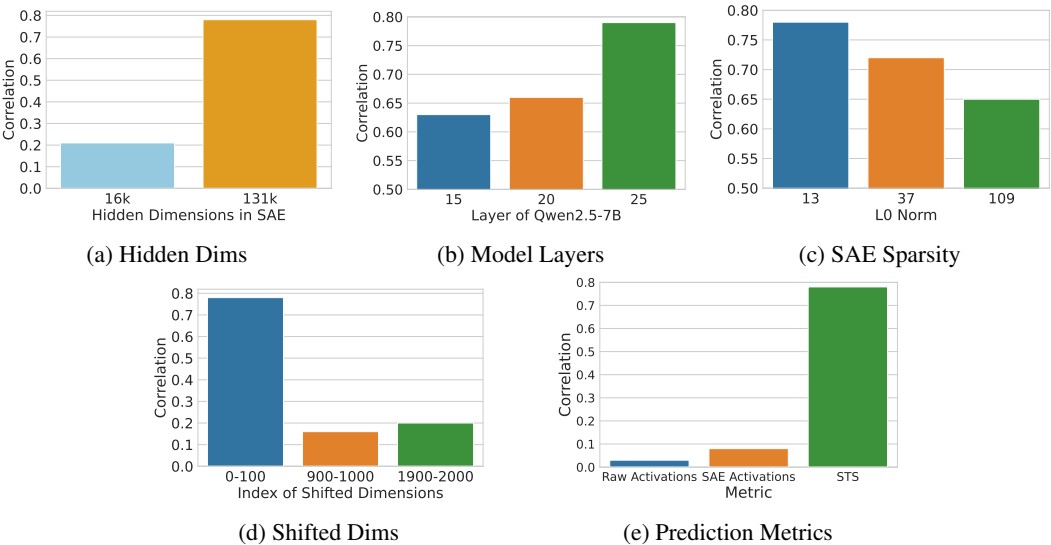

Figure 4: Ablation studies on the implementation of our metric. We evaluate (a) SAEs with varying hidden dimensions in the representation space, (b) SAEs trained on different layers of the pre-trained model, (c) different ranges of top-shifted dimensions, (d) different sparsity in SAE representations, e) the comparison between STS and directly using activations.

We conduct several ablation studies to evaluate STS under different conditions. First, we examine the role of monosemanticity by extracting SAE features with different hidden dimensions (16k vs. 131k). As shown in Figure 4a, weaker monosemanticity causes a clear drop in prediction accuracy, indicating its importance for STS. We also compare SAEs applied to different layers of Qwen2.5-7B-Instruct (layers 15, 20, and 25). Figure 4b shows that STS consistently correlates with performance changes across layers, confirming its robustness. Finally, we vary the sparsity levels of SAE representations. As illustrated in Figure 4c, higher sparsity—typically linked to stronger monosemanticity—yields more accurate predictions, further underscoring the essential role of monosemantic features in STS.

In addition to evaluating different SAEs, we examine another critical hyperparameter of our metric: the range of estimated shifted dimensions. Selecting dimensions with small shifts risks including those unaffected by the SFT process. As shown in Figure 4d, the correlation between STS and downstream performance decreases when smaller-shifted dimensions are selected, supporting this analysis. These findings underscore that the choice of shifted dimensions directly impacts metric reliability, highlighting the need for an appropriate selection strategy that only identify dimensions with largest shifts.

Furthermore, we conducted additional experiments to directly predict the performance improvements from SFT using model activations. In Figure 4e, we compare our method with predicting the improvements based on the model activations (using an optimized probe). As shown in the table, neither raw activations nor SAE feature activations exhibit meaningful correlation with the actual performance shifts. These results suggest that simply probing activations is insufficient; identifying the shifted dimensions induced by SFT is essential. Besides, we also note that this is a task where SAE beats probes, which further shows the potential of the monosemantic representation space.

## 5    APPLICATIONS: A DATA MIXTURE PRINCIPLE

In Section 4, we demonstrated that our proposed metric, the SAE-based transferability score (STS), exhibits a strong correlation with actual performance changes. Building on this result, we now explore a practical application of STS. Specifically, we leverage predicted transferability to optimize data mixture strategies during post-training. Using STS, we can identify the domains most likely to be affected by supervised fine-tuning (SFT). A common approach to mitigate performance degradation in such domains is to introduce additional data. Intuitively, with a predicted ranking of

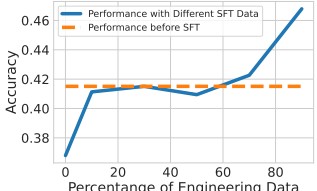 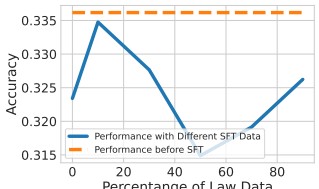 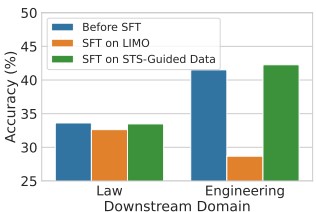

(a) Performance with Different Amounts of Engineering Data

(b) Performance with Different Amounts of Law Data

(c) Performance Shifts with Different SFT Data

Figure 5: Comparison of data mixture strategies in the SFT process. We focus on the domains with the largest (engineering) and smallest (law) performance shifts induced by SFT of Qwen2.5-7B-Instruct on LIMO. In total, 220 extra examples from a mixture of engineering and law data are added. Figure (a) reports engineering performance with varying amounts of engineering data, while Figure (b) reports law performance with varying amounts of law data. Figure (c) compares the downstream performance without additional data and with additional data mixed according to the ratio of their corresponding STS values.

performance changes across domains, we can allocate more data to those at higher risk of degradation. To validate this idea, we examine two domains in MMLU-Pro: engineering (with the largest performance drop when fine-tuning Qwen2.5-7B-Instruct on LIMO) and law (with the smallest). In our experiments, we fine-tune Qwen2.5-7B-Instruct on LIMO with additional data in MMLU-Pro. We split MMLU-Pro into training and testing sets with a 1:1 ratio. For training, we use Qwen2.5-7B-Instruct's outputs prior to SFT as supervised answers. In addition to the original LIMO data, we augment the training set with 220 extra examples sampled from the mixture of engineering and law domains.

As shown in Figure 5, we observe that domains with larger performance changes require more additional data. For example, allocating more data to the engineering domain leads to substantial improvements (Figure 5a), whereas allocating extra data to the law domain yields only marginal gains (Figure 5b). Moreover, when the data mixture ratio is adjusted to align with the ratio of their corresponding STS values, the resulting model achieves balanced performance across both engineering and law (Figure 5c). These findings suggest that STS can serve as an interpretable guide for designing data mixture ratios, enabling more effective post-training while mitigating uneven performance shifts across domains.

## 6 EXPLORATIONS ON REINFORCEMENT LEARNING

In this paper, we primarily investigate the impact of supervised fine-tuning across different domains. Nevertheless, reinforcement learning (RL) represents another non-negligible post-training paradigm, motivating us to explore whether our method can be extended to RL. We begin by applying the STS metric directly, as in the supervised fine-tuning setting. Specifically, we train Qwen2.5-7B-Instruct using the GRPO framework (Shao et al., 2024) on the Math-LightEval dataset and evaluate performance changes across domains in MMLU-Pro. When computing STS, we follow the same procedure as in supervised fine-tuning: ground-truth CoTs from Math-LightEval serve as demonstrations to estimate shifted dimensions, and correlations between these dimensions and downstream domains are calculated based on in-context learning. However, as shown in Figure 6b, STS exhibits low correlations with performance changes in the RL setting. In the following, we try to find the reasons behind this discrepancy.

We note that a key distinction between SFT and RL is that RL lacks access to ground-truth answers, making it challenging to select appropriate demonstrations for in-context learning. As a result, the estimation of shifted features may be inaccurate. To test this hypothesis, we compare the overlaps between the actual and predicted shifted dimensions. As shown in Figure 6a, the overlap is substantially lower in RL than in SFT. To further validate this observation, we replace the estimated top-100 dimensions with the ground-truth dimensions with the largest changes after RL and recompute the STS metric. Figure 6b demonstrates that STS calculated with ground-truth shifted dimensions shows a strong correlation with performance changes, indicating that the main challenge lies in accurately

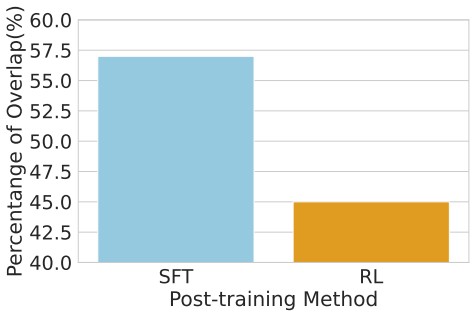
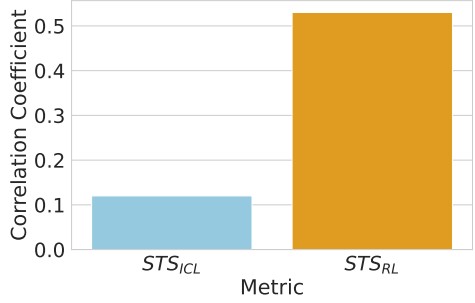

(a) Overlaps between Estimated and Actual Shifted Dimensions Induced by SFT and RL

(b) Different Implementations of STS Metric

Figure 6: The explorations on extending STS metric to the RL scenarios. We optimize Qwen2.5-7B-Instruct on Math-LightEval using GRPO. Figure (a) shows that the identified dimensions in RL show smaller overlap with the actual shifted dimensions compared to SFT, which limits the accuracy of the metric. Figure (b) shows that when the estimated shifted dimensions are replaced with the actual ones, the correlation between STS and downstream performance changes becomes much stronger.

estimating shifted dimensions in RL prior to training. And this will be the future direction of our explorations. For now, STS in RL can serve as a metric to predict transferability without evaluating RL models on downstream tasks.

## 7 CONCLUSION

In this work, we introduce a metric for predicting the transferability of post-training in large-scale neural networks, leveraging the interpretability of sparse autoencoders (SAEs). By identifying the SAE dimensions that are shifted during post-training and measuring their correlations with downstream domains, we propose the SAE-based transferability score (STS) as a predictive indicator of performance changes. Our experiments show that STS reliably forecasts performance variations across multiple domains, providing new insights into the internal mechanisms of post-training. Beyond supervised fine-tuning, we further demonstrate the applicability of our approach to reinforcement learning settings. Overall, we believe our work establishes an interpretable framework for understanding and anticipating post-training effects, paving the way for more targeted and effective post-training strategies.

## ACKNOWLEDGEMENT

Yisen Wang is supported by National Natural Science Foundation of China (62376010, 92370129), Beijing Major Science and Technology Project under Contract no. Z251100008425006, Beijing Nova Program (20230484344, 20240484642), and State Key Laboratory of General Artificial Intelligence.

## ETHICS STATEMENT

This work makes use of publicly available datasets and models. No private or sensitive data are involved, and no harmful content is included. Therefore, we believe this paper does not raise any ethical concerns.

## REPRODUCIBILITY STATEMENT

To ensure the reproducibility of our results, we provide comprehensive details of our experiments in both the main paper and the appendix. In Section 3, we describe the details of the methodology for identifying shifted dimensions, while Section 4 presents the procedure for computing correlations

between these dimensions and downstream domains. Additionally, Appendix A reports the concrete performance changes across different MMLU-Pro domains, and Appendix B offers further details and complementary discussions.

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

# A  PERFORMANCE CHANGES AFTER SFT

Table 2: Performance change on different domains of MMLU-Pro (Llama3-8B-Instruct fine-tuned on LIMO).

| | engineering | physics | chemistry | law |
|---|---|---|---|---|
| Performance Change (%) | -11.97 | -5.7 | -5.39 | -3.72 |
| | philosophy | other | health | computer science |
| Performance Change (%) | -2.61 | -2.49 | -1.1 | -0.97 |
| | economics | math | psychology | biology |
| Performance Change (%) | -0.95 | -0.89 | -0.63 | -0.42 |
| | history | business | | |
| Performance Change (%) | -0.26 | -0.12 | | |

Table 3: Performance change on different domains of MMLU-Pro (Qwen2.5-7B-Instruct fine-tuned on LIMO).

| | engineering | chemistry | physics | computer science |
|---|---|---|---|---|
| Performance Change (%) | -9.8 | -6.36 | -5.24 | -4.88 |
| | business | health | math | economics |
| Performance Change (%) | -4.43 | -2.93 | -2.89 | -2.61 |
| | philosophy | biology | other | psychology |
| Performance Change (%) | -2.4 | +1.53 | -1.3 | -0.37 |
| | history | law | | |
| Performance Change (%) | +0.26 | -0.19 | | |

Table 4: Performance change on different domains of MMLU-Pro (Gemma2-9B-Instruct fine-tuned on LIMO).

| | engineering | computer science | other | law |
|---|---|---|---|---|
| Performance Change (%) | -9.29 | -3.66 | -2.05 | -1.73 |
| | health | math | business | history |
| Performance Change (%) | -1.47 | +1.41 | -1.27 | -0.79 |
| | economics | chemistrcy | biology | physics |
| Performance Change (%) | -0.71 | -0.62 | -0.56 | -0.54 |
| | psychology | philosophy | | |
| Performance Change (%) | -0.05 | 0 | | |

We show the concrete signed performance change of models after SFT on LIMO in Table 2,3, 4. In our experiments with SFT on LIMO, we observe that performance decreases across nearly all downstream domains, and the primary difference between domains lies in the magnitude of the decrease. This is consistent with the known limitations of SFT in generalization. Consequently, our work focuses on predicting the magnitude of performance change rather than its sign. We consider this meaningful because accurately estimating the degree of decrease provides insights into model behavior under SFT and informs strategies to mitigate these decreases (e.g., the STS-guided data mixing strategy in Section 5).

## B EXPERIMENTS DETAILS

### B.1 DETAILS OF APPLIED SAEs

For feature extraction with sparse autoencoders (SAEs), we use one SAE for each backbone model on a specific layer of the transformer. Each SAE is an encoder-decoder architecture. The encoder and decoder are two linear layers while there exists an activation function following the encoder. We introduce the details of SAEs in the following.

Concretely, for Llama3-8B-Instruct, we use a ReLU SAE with 16384 hidden dimensions trained on residual-stream activations from layer 25. The SAE is trained on the openWebText dataset with context size as 1024. The SAE is optimized using the AdamW optimizer with $\beta_1$=0.9, $\beta_2$=0.999, and weight decay of 0.01. The learning rate is set to 1e-5. An L1 sparsity penalty of 5 (with warm-up) is applied on the hidden activations to induce sparse and monosemantic features, following standard SAE training practices.

For Qwen2.5-7B-Instruct, we use a ReLU SAE with 28672 hidden dimensions trained on residual-stream activations from layer 25. The SAE is trained on the openWebText dataset with a context size of 2048. The model is optimized using the AdamW optimizer with $\beta_1$=0.9, $\beta_2$=0.999, and a cosine-annealing learning-rate schedule starting at 7e-5 with warm-up. An L1 sparsity penalty of 5 (with warm-up) is applied to encourage sparse and monosemantic features, following standard SAE training setups.

For Gemma2-9B-Instruct, we use a ReLU SAE with 131072 hidden dimensions trained on residual-stream activations from layer 31. The SAE is trained on the openWebText dataset with a context size of 1024. The model is optimized using the AdamW optimizer with $\beta_1$=0.9, $\beta_2$=0.999, and a cosine-annealing learning-rate schedule over the first 10,000 steps. We also apply a linear warmup of the sparsity coefficient over the first 10,000 steps to stabilize training and encourage sparse feature activations.

### B.2 DETAILS OF ICL PROMPTS

To identify shifted SAE dimensions induced by post-training, we first sample activations from 20,000 tokens before and after in-context learning (ICL), where the ICL demonstrations are constructed using supervised answers from LIMO. The prompt looks likes [x1,y1,x2,y2,x3], where x1,x2,x3 are the questions in LIMO while y1, y2 are the responses in LIMO. To be specific, we provide a concrete example in the following

---

**In-context Learning Prompts**

[{'content': "A fenced, rectangular field measures 24 meters by 52 meters.

...
What is the largest number of square test plots into which the field can be partitioned using all or some of the 1994 meters of fence? Let's think step by step and output the final answer within \boxed{}.",
'role': 'user'},
{'content': 'Okay, so I have this problem where there´s a rectangular field that´s 24 meters by 52 meters. The farmer wants to partition this entire field into square test plots, with the sides of the squares parallel to the edges of the field.

...
So, I´ll call this over. Thus, the answer is 702.
**Final Answer**
\boxed{702}',
'role': 'assistant'},
{'content': "A hotel packed breakfast for each of three guests.

...
Given that the probability each guest got one roll of each type is $\frac{m}{n}$, where $m$ and $n$ are relatively prime integers, find $m + n$. Let's think step by step and output the final answer within \boxed{}.",

---

```
'role': 'user'},
{'content': "Okay, so here's this problem about a hotel packing breakfast for three guests.
Each breakfast is supposed to have one nut roll, one cheese roll, and one fruit roll.
...
The total number of ways to choose three rolls from the remaining 6: C(6,3)=20. So proba-
bility Therefore, 9/70 is correct. Thus, m + n=79. Therefore, the answer is 79.
**Final Answer**
\boxed{79}",
'role': 'assistant'},
{'content': "For how many pairs of consecutive integers in
1000, 1001, 1002, . . . , 2000 is no carrying required when the two integers are added? Let's
think step by step and output the final answer within \boxed{}.",
'role': 'user'}]
```

We then compute the activation differences across SAE dimensions and select those exhibiting the largest shifts. Finally, to quantify the correlations between shifted dimensions and downstream domains, we sample activations from 10,000 tokens for each domain and compute their correlations with the identified dimensions.

### B.3 DETAILS OF SUPERVISED FINE-TUNING

When fine-tuning the pre-trained models Qwen2.5-7B-Instruct, Llama3-8B-Instruct, and Gemma2-9B-Instruct on the LIMO dataset, we adopt a unified experimental setup across models. Specifically, the maximum prompt length is set to 8192 tokens, with sequences truncated from the left to fit within this constraint. All models are fine-tuned for 10 epochs using four H20 GPUs. We train the models with a total batch size of 256 and a micro-batch size of 1 per GPU. Models are trained using FSDP with fp32 precision, gradient checkpointing enabled, and no CPU offload. We use the AdamW optimizer with betas (0.9, 0.95), weight decay 0.01, and gradient clipping of 1.0. A cosine learning rate scheduler is applied with a warmup of 10% of the total training steps.

## C ADDITIONAL EXPERIMENTS

### C.1 COMPARISON WITH TRADITIONAL REPRESENTATION SHIFT ANALYSIS

We note that our work is not a trivial reframe of existing analyses on representation drift or feature correlations. The key distinction is that prior studies examine raw representation shifts **after** fine-tuning, whereas our paper demonstrates that **the sparsity and monosemanticity brought by SAEs enable us to predict feature shifts and find correlations before fine-tuning**. As shown in Figure 2, when using raw model features without SAEs, the overlap between predicted and actual shifted dimensions is quite low, indicating that **traditional representation analyses cannot accurately identify shifted dimensions**.

To further distinguish our method from traditional representation analysis, we conduct additional experiments comparing STS with three baselines: (1) raw feature activations in downstream domains, (2) representation similarity between downstream and training domains, and (3) representation similarity between models before and after SFT. For Qwen2.5-7B-Instruct, Table 1 reports the correlations between these measures and actual performance shifts.

Table 5: Correlation coefficient between actual performance shifts (Qwen2.5-7B-Instruct tuned on LIMO) and different baselines.

|  | Feature Activations | Representation Similarity between Downstream Domains and Training Domain | Representation Similarity between Models before and after SFT | STS |
|---|---|---|---|---|
| Uses Model after SFT | No | No | Yes | **No** |
| Coefficient | 0.03 | 0.11 | 0.61 | **0.79** |

As shown in the table, neither raw feature activations nor representation similarity between training and downstream domains strongly correlates with performance shifts. Even when using post-SFT representations, the correlation remains substantially lower than that of our method. These results further demonstrate that our approach is not a minor variation of traditional representation drift analyses.

## C.2    REPEATED EXPERIMENTS ON EVALUATING THE CORRELATION COEFFICIENT

We ran additional experiments with three independent seeds and report the mean ± standard deviation in the table below.

Table 6: The Pearson correlation coefficient between STS and actual performance shifts on MMLU-Pro induced by SFT on LIMO.

| Metric / Model | LLaMA3-8B | Qwen2.5-7B | Gemma2-9B |
|---|---|---|---|
| $STS_{act}$ | $0.71 \pm 0.01$ | $0.90 \pm 0.02$ | $0.60 \pm 0.03$ |
| $STS_{ICL}$ | $0.81 \pm 0.01$ | $0.78 \pm 0.01$ | $0.77 \pm 0.01$ |

As shown in the table, the results further verify that the correlations are statistically significant.

## C.3    QUALITATIVE ANALYSIS OF THE IDENTIFIED SHIFTED DIMENSIONS

We annotate SAE dimensions following the auto-interpretability scoring pipeline in (Cunningham et al., 2023). The procedure is as follows:

1.  We construct a dataset consisting of three domains: math (LIMO), code (Verifiable-Coding-Problems-Python-10k-Dataset), and dialogue (HH-RLHF).

2.  We encode these samples and extract the corresponding SAE features (using the 25th layer of Qwen2.5-7B-Instruct as an example) . 3. For each SAE dimension, we collect the top 10 samples with the highest activations.

4.  We then prompt an LLM (Llama3-8B-Instruct) to determine whether these samples belong to math, code, or general dialogue.

5. Finally, each dimension is assigned a label (math/code/dialogue) based on the LLM's judgment.

With the annotated data, we respectively calculate whether the top 50, top 100, and top 200 estimated shifted dimensions belong to the training task, i.e., the math.

Table 7: The percentage of the estimated shifted dimensions that are explained as the math dimension.

| Selected Dimensions | 50 | 100 | 200 |
|---|---|---|---|
| | 89% | 93% | 92% |

As shown in the table, the estimated shifted dimensions show an extremely high correlation with the training task (math), which further verifies the effectiveness of our method.

Furthermore, we then evaluate how many math-related features are recalled among the top-changed SAE activations. Specifically, we sample 100 dimensions annotated as math and 100 dimensions annotated as code. We then compute the proportion of these dimensions that appear among the top 500 estimated shifted dimensions.

As shown in the table, math-related dimensions are recalled at a substantially higher rate than code-related dimensions. This demonstrates that our estimation process accurately identifies the SAE dimensions most relevant to the training task.

Table 8: Proportion of 100 annotated dimensions recalled among the top 500 estimated shifted dimensions.

| Annotation | Math | Code |
|---|---|---|
| Recall | 63% | 7% |

## C.4 R2 between STS and Actual Performance Shifts

We report the corresponding R² results in the following table.

Table 9: R² between STS and actual performance shifts on MMLU-Pro induced by SFT on LIMO.

| Metric / Model | LLaMA3-8B | Qwen2.5-7B | Gemma2-9B |
|---|---|---|---|
| $STS_{act}$ | $0.50 \pm 0.01$ | $0.80 \pm 0.04$ | $0.36 \pm 0.03$ |
| $STS_{ICL}$ | $0.66 \pm 0.01$ | $0.61 \pm 0.02$ | $0.60 \pm 0.02$ |

As shown in the table, both $STS_{act}$ and $STS_{ICL}$ are effective across models. Notably, the $STS_{ICL}$ values are highly consistent across the three models (0.60–0.66 with small standard deviations), underscoring the effectiveness of this metric in predicting performance shifts.

## C.5 Test Accuracy on the Math Dataset

Table 10: Test accuracy (%) of Qwen2.5-7B-Instruct before and after SFT on LIMO.

| Before SFT | SFT on LIMO | SFT on STS-Guided Data |
|---|---|---|
| 74.7 | 81.3 (+6.6) | 80.1 (+5.4) |

In Figure 5, we report downstream accuracy on the law and engineering domains. **It is important to note that the SFT process was conducted on the math dataset LIMO.** In the following table, we present the test accuracy on the Math-LightEVAL dataset before and after SFT.

As shown in the table, SFT leads to a substantial improvement on Math-LightEVAL. Combined with the observations in Figure 5, these results indicate that SFT on STS-guided data effectively enhances math performance while maintaining the model's capabilities in other domains.

## C.6 Selectively Fine-tuning on Estimated Shifted Dimensions

Table 11: Test accuracy (%) of Qwen2.5-7B-Instruct after selectively fine-tuning.

| Selected Dims | Before SFT | SFT on Random Dimensions | SFT on Estimated Shifted Dimensions |
|---|---|---|---|
| 1000 | 74.7 | 74.2 | 75.6 |
| 3000 | 74.7 | 74.0 | **76.9** |

To further evaluate whether the selected dimensions are the most relevant to the training task, we conduct selectively fine-tuning based on identified shifted dimensions. To be specific, we first selected five layers in Qwen2.5-7B-Instruct/Llama3-8B-Instruct and extracted 3000 dimensions from the SAE representation space at each layer. We then added five linear layers of shape [3000, $d$], where $d$ is the dimension of the raw (pre-SAE) features. During the forward pass, the 3000 SAE dimensions are decoded through these learnable linear layers and added back to the raw features. We fine-tune only these five linear layers on LIMO, keeping all other model parameters frozen. As shown in Figure 7, selective fine-tuning using the estimated shifted dimensions effectively improves math performance with only five linear layers. This result further validates that the selected dimensions are the most relevant to the training domain.

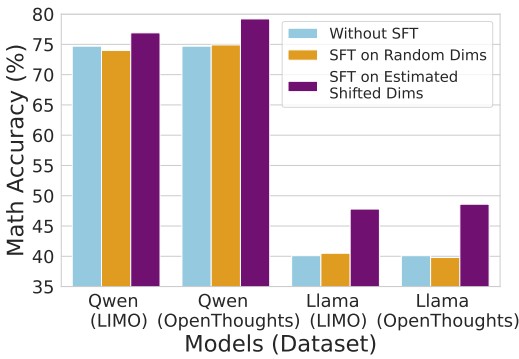

Figure 7: Selective Finetuning Different Models on Estimated Shifted Dimensions

Indeed, in our selective fine-tuning experiments, we tune the top-$K$ dimensions rather than a single dimension ($K = 3000$ in Table 11). We also evaluate $K = 1000$, with the results shown below.

As shown in the table, selective fine-tuning on the estimated 1000 or 3000 shifted dimensions effectively improves math performance. These results further confirm that the identified shifted dimensions are highly aligned with the training task.

## C.7 ABLATION STUDY ON THE SPARISTY OF SAEs

Following prior work, we experiment with different sparsity levels in this representation space. The following Table reports the results using SAEs trained with varying L0 norms.

Table 12: Pearson correlation coefficients between STS and actual performance shifts on MMLU-Pro induced by fine-tuning Gemma2-9B-Instruct on LIMO. STS is computed using SAEs with different $L_0$ norms.

| $L_0$ | 13 | 22 | 37 | 63 | 109 |
|---|---|---|---|---|---|
| Correlation | 0.78 | 0.77 | 0.72 | 0.75 | 0.65 |

As shown in the table, increased sparsity in the SAE leads to more accurate prediction of performance changes. Since stronger sparsity is generally associated with stronger monosemanticity (Cunningham et al., 2023), these results further highlight the critical role of monosemantic representations in our method.

## C.8 PREDICTING PERFORMANCE SHIFTS BASED ON MODEL ACTIVATIONS

We conducted additional experiments to directly predict the performance improvements from SFT using model activations. Using Qwen2.5-7B-Instruct as an example, we fine-tune the model on LIMO and measure the performance shifts across different MMLU-Pro domains. In the following table, we compare our method with predicting the improvements based on the model activations (using an optimized probe).

Table 13: Comparison between STS and predictions based on model activations.

| | Raw Feature Activations | SAE Feature Activations | STS |
|---|---|---|---|
| Correlation Coefficient | 0.03 | 0.08 | **0.79** |

As shown in the table, neither raw activations nor SAE feature activations exhibit meaningful correlation with the actual performance shifts. These results suggest that simply probing activations is insufficient; identifying the shifted dimensions induced by SFT is essential for understanding and predicting model behavior during the fine-tuning process.

### C.9 Additional Discussion on Figure 2b

As shown in Figure 2(b) of the paper, we compare the raw and SAE dimensions in terms of the proportion of total shifts captured by the top shifted dimensions. We observe that the top 1% of raw features account for a smaller fraction, indicating that the shifts are distributed relatively uniformly across the raw dimensions, which makes it difficult to identify the core shifted features. In contrast, shifts in the SAE dimensions are more concentrated, highlighting the effectiveness in capturing key shifted dimensions.

### C.10 Additional Discussion on the Monosemanticity Assumption

We note that recent empirical works and theoretical studies (Cunningham et al., 2023; Gao et al., 2024; Cui et al., 2025) provide consistent evidence that sparse autoencoders trained on LLMs tend to yield monosemantic representations. These findings support the validity of relying on the monosemanticity assumption in our method. We also believe that the monosemantic representation induced by SAEs is a key distinction of our approach compared with traditional representation analysis techniques because it allows us to identify shifted features in a semantically interpretable space and thereby predict transferability before conducting SFT. Besides, we clarify that our method does not require any additional curated demonstrations beyond the responses already used for SFT. In practice, we only randomly sample two responses from the SFT training set.

## D The Use of Large Language Models (LLMs)

In this work, the use of LLMs was limited to minor language editing to improve readability. All conceptual development, theoretical analysis, experimental design, and result interpretation were conducted independently by the authors. Thus, the use of LLMs was purely auxiliary and had no impact on the scientific contributions of this paper.

