# OpenReview forum: "SAE as a Crystal Ball: Interpretable Features Predict Cross-domain Transferability of LLMs without Training"
_ICLR.cc/2026/Conference — ICLR 2026 Poster_

### Official Review · Reviewer_MqTi · 2025-10-17

**Soundness:** 2
**Presentation:** 3
**Contribution:** 3
**Rating:** 4
**Confidence:** 3

**Summary:**

This paper shows that SAE activations on specific domains (in ICL) can predict gains from SFT on the same domain, with high correlation.
It studies what portion of features that are activated during ICL also get bolstered by SFF, and it then does ablation studies to check their impact on model accuracy. It checks the portion of overlapping features between ICL and SFT, and it makes sure that results hold in RL (with a new formulation). The paper finally proposes an application to designing the data mixture used for SFT based on SAE feature activations).

**Strengths:**

The main strengths of this papers lie in the novel insight that SAE activations in ICL can predict improvements from SFT. The link between ICL and SFT was present in the literature, and the authors locate themselves in the literature well.
The paper is comprehensive, and it covers different modalities (RL/SFT) and applications (data mix for SFT). It correlates features with SFT gains well (0.7/0.8 correlation), and it shows the (very strong) effects of ablating features.
The paper also studies overlaps and uncovers significant portion of feature shifts (increase in activation under ICL) are the same as what SFT causes.

**Weaknesses:**

The main weaknesses are:
- no sample standard deviations for statistics you calculate
- no R2 for Figure 4's linear regressions, it would be interesting to know the fraction of the improvement from SFT you predict
- the SFT you do is can be ineffective (Figure 5), so predicting it is less interesting
- SAEs are lossy, and so is your predictive method (compounded lossyness)

I am open to increasing my score if the (first 2 or 3) weaknesses above are addressed

Minor:
- the contributions in the intro are too verbose

**Questions:**

Can you unpack the most important features in terms of neurons (that you could later fine tune selectively doing something like a selective LoRA)? Have you tried a sparsity penalty on SAE weights (to prune weights to save compute)?

Have you tried computing sample standard deviations and R2s?

How do you perform compared to trying to predict improvements form SFT based on model activations directly (using a probe)?

---

> ### Author Response · Authors · 2025-11-21
> **Response to Reviewer MqTi (1/2)**
>
> Thanks for your careful reading and critical review. Following your suggestions, we have added more experiments on computing the standard deviations and R2s. We further address each of your concerns below, and hope you find them satisfactory.
>
> ---
>
> **Q1.** No sample standard deviations for the statistics you calculate
>
> **A1.** Thanks for your suggestions! We ran additional experiments with three independent seeds and report the mean ± standard deviation in the table below.
>
>  *Table R1. The Pearson correlation coefficient between STS and actual performance shifts on MMLU-Pro induced by SFT on LIMO.*
>
> | Metric / Model | LLaMA3-8B | Qwen2.5-7B | Gemma2-9B |
> | --- | --- | --- | --- |
> | STS_act | 0.71 ± 0.01 | 0.90 ± 0.02 | 0.60 ± 0.03 |
> | STS_ICL | 0.81 ± 0.01 | 0.78 ± 0.01 | 0.77 ± 0.01 |
>
> These results further confirm that the observed correlations are statistically significant. We have added these results in Appendix C.3 of the revised version.
>
> ---
>
> **Q2.** No R2 for Figure 4's linear regressions. It would be interesting to know the fraction of the improvement from SFT you predict
>
> **A2.** Thanks for your suggestions. We report the corresponding R² results in the following table.
>
> *Table R2. R² between STS and actual performance shifts on MMLU-Pro induced by SFT on LIMO.*
>
> | Metric / Model | LLaMA3-8B | Qwen2.5-7B | Gemma2-9B |
> | --- | --- | --- | --- |
> | STS_act | 0.50 ± 0.01 | 0.80 ± 0.04 | 0.36 ± 0.03 |
> | STS_ICL | 0.66 ± 0.01 | 0.61 ± 0.02 | 0.60 ± 0.02 |
>
> As shown in the table, both STS_act and STS_ICL are effective across models. Notably, the STS_ICL values are highly consistent across the three models (0.60–0.66 with small standard deviations), underscoring the effectiveness of this metric in predicting performance shifts. We have added the results in Appendix C.5 of the revised version.
>
> ---
>
> **Q3.** The SFT you do is can be ineffective (Figure 5), so predicting it is less interesting
>
> **A3.** We believe there may exist a misunderstanding here. In Figure 5 of the paper, we report downstream accuracy on the law and engineering domains. **It is important to note that the SFT process was conducted on the math dataset LIMO.** In the following table, we present the test accuracy on the Math-LightEVAL dataset before and after SFT.
>
> *Table R3. Test accuracy (%) of Qwen2.5-7B-Instruct before and after SFT on LIMO.*
>
> | Before SFT | SFT on LIMO | SFT on STS-Guided Data |
> | --- | --- | --- |
> | 74.7 | 81.3 (+6.6) | 80.1 (+5.4) |
>
> As shown in the table, SFT leads to a substantial improvement on Math-LightEVAL. Combined with the observations in Figure 5 of the paper, these results indicate that SFT on STS-guided data effectively enhances math performance while maintaining the model’s capabilities in other domains. We have added these results in Appendix C.6 of the revised version.
>
> ---
>
> **Q4.** The contributions in the intro are too verbose
>
> **A4.** Thanks for your suggestions! We have revised the contributions part in the revised version to make it more concise. The revised version is as follows:
>
> - We propose a method to identify shifted dimensions in supervised fine-tuning without requiring access to the fine-tuned models. We observe that when supervised answers are used as context prompts, the shifted dimensions in in-context learning substantially overlap with those in supervised fine-tuning.
> - We propose the SAE-based Transferability Score (STS), which uses correlations in SAE feature space and estimated shifted dimensions to accurately predict LLM transferability without performing supervised fine-tuning.
> - We empirically show that higher STS values strongly correlate with larger performance shifts in supervised fine-tuning, achieving Pearson correlations above 0.75 across diverse scenarios. This confirms that STS is a reliable, fine-tuning-free metric for predicting LLM cross-domain transferability.

---

> ### Author Response · Authors · 2025-11-21
> **Response to Reviewer MqTi (2/2)**
>
> **Q5.**  Can you unpack the most important features in terms of neurons (that you could later fine tune selectively doing something like a selective LoRA)?
>
> **A5.** Indeed, selectively fine-tuning based on identified shifted dimensions is an interesting idea. To evaluate this idea, we conducted the following experiment. We first selected five layers in Qwen2.5-7B-Instruct and extracted 3000 dimensions from the SAE representation space at each layer. We then added five linear layers of shape [3000, d], where d is the dimension of the raw (pre-SAE) features. During the forward pass, the 3000 SAE dimensions are decoded through these learnable linear layers and added back to the raw features. We fine-tune only these five linear layers on LIMO, keeping all other model parameters frozen.
>
> *Table R4. Test accuracy (%) of Qwen2.5-7B-Instruct after selectively fine-tuning.*
>
> | Before SFT | SFT on Random dimensions | SFT on Estimated Shifted Dimensions |
> | --- | --- | --- |
> | 74.7 | 74.0 | **76.9** |
>
> As shown in the table above, selective fine-tuning using the estimated shifted dimensions effectively improves math performance with only five linear layers. This result further validates that the selected dimensions are the most relevant to the training domain. We have added these results in Appendix C.7 of the revised version.
>
> ---
>
> **Q6.** Have you tried a sparsity penalty on SAE weights (to prune weights to save compute)?
>
> **A6.** Indeed, we note that sparsity in SAEs is typically introduced through a sparsity penalty or by enforcing sparse activations in the SAE representation space. Following prior work, we experiment with different sparsity levels in this representation space. The following table reports the results using SAEs trained with varying L0 norms.
>
> *Table R5. Pearson correlation coefficients between STS and actual performance shifts on MMLU-Pro induced by fine-tuning Gemma2-9B-Instruct on LIMO. STS is computed using SAEs with different L0 norms.*
>
> | L_0 | 13 | 22 | 37 | 63 | 109 |
> | --- | --- | --- | --- | --- | --- |
> | Coefficient | 0.78 | 0.77 | 0.72 | 0.75 | 0.65 |
>
> As shown in the table, increased sparsity in the SAE leads to more accurate prediction of performance changes. Since stronger sparsity is generally associated with stronger monosemanticity [1], these results further highlight the critical role of monosemantic representations in our method. We have added these results in Appendix C.8 of the revised version.
>
> [1] Cunningham, Hoagy, et al. "Sparse autoencoders find highly interpretable features in language models." *arXiv preprint arXiv:2309.08600* (2023).
>
> ---
>
> **Q7.** Have you tried computing sample standard deviations and R2s?
>
> **A7.** Thanks for pointing it out, and we have reported these results in Q1 and Q2.
>
> ---
>
> **Q8.** How do you perform compared to trying to predict improvements form SFT based on model activations directly (using a probe)?
>
> **A8.** Thanks for your suggestion. We conducted additional experiments to directly predict the performance improvements from SFT using model activations. Using Qwen2.5-7B-Instruct as an example, we fine-tune the model on LIMO and measure the performance shifts across different MMLU-Pro domains. In the following table, we compare our method with predicting the improvements based on the model activations (using an optimized probe).
>
> *Table R6. Comparison between STS_ICL and predictions based on model activations.*
>
> |  | Raw Feature Activations | SAE Feature Activations | STS_ICL |
> | --- | --- | --- | --- |
> | Correlation Coefficient | 0.03 | 0.08 | **0.79** |
>
> As shown in the table, neither raw activations nor SAE feature activations exhibit meaningful correlation with the actual performance shifts. These results suggest that simply probing activations is insufficient; identifying the shifted dimensions induced by SFT is essential for understanding and predicting model behavior during the fine-tuning process. We have added the results in Appendix C.9 of the revised version.
>
> ---
>
> Thank you again for your encouraging comments and valuable feedback. We would appreciate if you could re-evaluate given these clarifications and enhancements. We are very happy to address your remaining concerns on our work.

---

> > ### Comment · Reviewer_MqTi · 2025-11-21
> >
> > Thanks for the additional experiments, they strengthen the paper a lot.
> >
> > I’ll increase my score to a 6.
> >
> > Remaining issues:
> > - why does your method appear not to work on qwen 2.5b
> > - move the new results to the main paper not appendices (good practice to have confidence intervals and ablations)
> > - highlight that this is a task where you beat probes (this blog post claims saes can’t: https://deepmindsafetyresearch.medium.com/negative-results-for-sparse-autoencoders-on-downstream-tasks-and-deprioritising-sae-research-6cadcfc125b9)

---

> > > ### Author Response · Authors · 2025-11-21
> > > **Further Response to Reviewer MqTi**
> > >
> > > Thanks for your responses and for improving the scores! Below, we will address your remaining concerns point by point.
> > >
> > > ---
> > >
> > > Q1. Why does your method appear not to work on qwen 2.5b
> > >
> > > A1. We would like to clarify that our paper does not report results on Qwen-2.5B. All Qwen experiments are conducted on the Qwen2.5-7B-Instruct model.  In addition,  we think the evaluations on Qwen2.5-7B-Instruct show stable and consistent performance; therefore, **our method does not appear to fail on Qwen models.** Therefore, the comment regarding “not working on Qwen-2.5B” may stem from a misunderstanding of the model used.
> > >
> > > ---
> > >
> > > Q2. Move the new results to the main paper not appendices (good practice to have confidence intervals and ablations)
> > >
> > > A2. Thank you for your suggestions! We have revised the paper and incorporated most of the new results into the main text, including standard deviation results (Figure 3 in the revised version), the ablation study on SAE sparsity levels (Figure 4(c) in the revised version), and the comparison with directly using activations (Figure 4(e) in the revised version). Please let us know if there are any additional results you believe should appear in the main paper.
> > >
> > > ---
> > >
> > > Q3. Highlight that this is a task where you beat probes
> > >
> > > A3. Thank you for your suggestions! We have added a discussion comparing STS with directly predicting transferability using probes in the revised paper (Lines 400-406), and we highlight that this is a task where SAEs beat probes.
> > >
> > > ---
> > >
> > > Thanks again for your constructive and detailed comments. We are very happy to address the remaining concerns about our paper.

---

> > > > ### Comment · Reviewer_MqTi · 2025-11-21
> > > >
> > > > it seems to me STS_act > STS_ICL for all Qwen results in your comments above, how does that imply that your method works on qwen?

---

> ### Author Response · Authors · 2025-11-21
> **Further Response to Reviewer MqTi**
>
> Thank you for your comment. Indeed, STS_act and STS_ICL are two variants of our method. Both follow the same pipeline: we first identify the shifted SAE dimensions using activation changes before and after ICL on the training set, and then compute their correlation with downstream domains using either (i) SAE activations in these estimated shifted dimensions (STS_act) or (ii) SAE activation changes before and after ICL in these dimensions (STS_ICL). The difference between the two scores does not indicate that the method fails on Qwen; rather, it reflects two alternative implementations of the same framework. Moreover, since STS_ICL exhibits more consistent performance across the three evaluated models, we adopt STS_ICL for the ablation study.
>
> We hope this clarifies the distinction between STS_act and STS_ICL and addresses your concern. Thank you for your valuable feedback.

---

> > ### Comment · Reviewer_MqTi · 2025-11-21
> >
> > Given everything I think this could be a good paper, but I'd say the last thing missing for it to be a strong accept would be more comprehensive experiments of this kind:
> >
> > Table 4. Test accuracy (%) of Qwen2.5-7B-Instruct after selectively fine-tuning.
> >
> > Before SFT	SFT on Random dimensions	SFT on Estimated Shifted Dimensions
> > 74.7	74.0	76.9
> >
> >
> > so in particular more than just 1 dimension, but something like a top-K dimensions approach

---

> ### Author Response · Authors · 2025-11-21
> **Further Response to Reviewer MqTi**
>
> Thank you for your comment. Indeed, in our selective fine-tuning experiments, we tune the top-K dimensions rather than a single dimension (K=3000 in Table 4). We also evaluate K=1000, with the results shown below.
>
> *Table R4. Test accuracy (%) of Qwen2.5-7B-Instruct after selectively fine-tuning.*
>
> | Selected Dimensions  | Before SFT | SFT on Random dimensions | SFT on Estimated Shifted Dimensions |
> | --- | --- | --- | --- |
> | 1000 | 74.7 | 74.2 | 75.6 |
> | 3000 | 74.7 | 74.0 | **76.9** |
>
> As shown in the table, selective fine-tuning on the estimated 1000 or 3000 shifted dimensions effectively improves math performance. These results further confirm that the identified shifted dimensions are highly aligned with the training task.
>
> We are wondering if our understanding correctly captures the point you raised? If more experiments are desired, we are happy to explore additional selections of dimensions and other models.

---

> > ### Comment · Reviewer_MqTi · 2025-11-21
> >
> > This is the table 4 in the paper
> >
> > Table 4: Correlation coefficient between actual performance shifts (Qwen2.5-7B-Instruct tuned on
> > LIMO) and different baselines.

---

> > > ### Author Response · Authors · 2025-11-21
> > > **Further Response to Reviewer MqTi**
> > >
> > > We noticed there may be some confusion regarding table numbering between the main paper and the rebuttal. Now, in the rebuttal, we re-label tables as R1, R2, etc. Specifically, Table R4 shows the test accuracy (%) of Qwen2.5-7B-Instruct after selective fine-tuning, while Table 4 in the main paper reports the correlation coefficient between actual performance shifts (Qwen2.5-7B-Instruct tuned on LIMO) and different baselines. Could you clarify what concern you would like us to address regarding these tables?

---

> > > > ### Comment · Reviewer_MqTi · 2025-11-21
> > > >
> > > > That’s not reflected in the revised version, also, I’d be great to see a couple more models in there / some other dataset

---

> ### Author Response · Authors · 2025-11-24
> **Further Response to Reviewer MqTi**
>
> Thanks for your comment. We have conducted additional experiments and added them into the revised version of the paper (**Figure 2(c)**). To be specific, we follow the default settings as fine-tuning Qwen2.5-7B-Instruct on LIMO and conduct selective fine-tuning on **another dataset (OpenThoughs—114K) and another model (LLama3-8B-Instruct)**. The results are summarized as follow.
>
> *Table R7. Test accuracy (%) after selectively fine-tuning on different models and different datasets.*
>
> |  | Before SFT | SFT on Random dimensions | SFT on Estimated Shifted Dimensions |
> | --- | --- | --- | --- |
> | Qwen on LIMO | 74.7 | 74.0 (-0.7) | 76.9 (+2.2) |
> | Qwen on Openthoughts | 74.7 | 74.9 (+0.2) | 79.2 (+4.5) |
> | Llama on LIMO | 40.1 | 40.5 (+0.4) | 47.8 (+7.7) |
> | Llama on OpenThoughts | 40.1 | 39.8 (-0.3) | 48.6 (+8.5) |
>
> As shown in the table, selective fine-tuning on the estimated shifted dimensions of different models consistently effectively improves math performance. These results further confirm that the identified shifted dimensions are highly aligned with the training task. We have added these results in Figure 2(c) of the revised version.
>
> Thanks again for your constructive comments. We are very happy to address the remaining concerns about our paper.

---

> > ### Comment · Reviewer_MqTi · 2025-11-25
> >
> > wow that's very good
> >
> > make sure to emphasise this fact in the camera ready a lot, and everything else we discussed
> >
> >
> > I'm raising to 8

---

### Official Review · Reviewer_efyn · 2025-10-27

**Soundness:** 2
**Presentation:** 3
**Contribution:** 2
**Rating:** 4
**Confidence:** 3

**Summary:**

This paper investigates the problem of the unpredictable transferability of post-training to downstream tasks. Post-training on data can introduce model shifts that improve the performance of some tasks while degrading others. To address this issue, the paper proposes a Sparse Autoencoder (SAE)-based Transferability Score (STS) to forecast post-training transferability. The method leverages supervised answers as demonstrations for in-context learning and identifies the SAE dimensions that exhibit the largest changes, which correlate with downstream task performance. Experiments across multiple models and domains demonstrate that the proposed transferability score accurately predicts the effects of both supervised fine-tuning and reinforcement learning (RL) tuning.

**Strengths:**

S1: The paper focuses on an interesting problem of forecasting changes in downstream task performance without additional training.

S2: The proposed fine-tuning-free approach for estimating transferability is both novel and useful.

S3: The investigation of SAE dimension shifts under both fine-tuning and in-context learning (ICL) is insightful.

S4: The paper evaluates the proposed STS method across multiple major open models (Qwen2.5-7B, Llama3-8B, Gemma2-9B) and demonstrates consistently high correlations.

**Weaknesses:**

W1: The study is limited to a single training dataset (LIMO) and a single evaluation benchmark (MMLU-Pro). Broader domains (e.g., dialogue, code generation) and larger model scales remain underexplored, even though these are areas where STS would likely be most valuable.

W2: There are potential reproducibility issues, as details on the SAE architectures, training procedures, hyperparameters, and prompt templates are either missing or insufficiently explained.

W3: Methodologically, STS relies heavily on the monosemanticity assumption of SAEs that each latent corresponds to a distinct human-interpretable concept. It also depends on access to high-quality demonstrations to estimate feature shifts (based on the weaker correlations observed in the RL experiments).

**Questions:**

Q1: Figure 2: What exactly do “raw model dimensions” refer to?

Q2: What does the ICL prompt template look like?

Q3: Line 243: The statement “the shifted features before SAE are more uniformly distributed” could be clarified. Does this mean less sparsity, or something else?

---

> ### Author Response · Authors · 2025-11-21
> **Response to Reviewer eyfn (1/2)**
>
> Thanks very much for your constructive and detailed comments. Below, we will address your main concerns point by point.
>
> ---
>
> **Q1.** Broader domains remain under-explored.
>
> **A1.** Thanks for your suggestions. We have conducted additional experiments across different training domains. Specifically, we include a code-generation dataset (Verifiable_Coding_Problems_Python_10k_Dataset) and a health dialogue dataset (CoT_Clinical_MH_Reasoning_Dataset) to further evaluate the correlation between ICL-induced feature drift and SFT-induced shifts. In addition, we examine how well our metric, STS_ICL, correlates with actual downstream performance changes. Taking Qwen2.5-7B-Instruct as an example, the empirical results are summarized in the following table.
>
> *Table R1. Verification on a code dataset (Verifiable_Coding_Problems_Python_10k_Dataset) and a clinical reasoning dataset (CoT_Clinical_MH_Reasoning_Dataset). The Qwen-2.5-7B-Instruct model is trained on each dataset, and we evaluate: (1) the correlation between ICL feature drift and SFT, and (2) the correlation between actual performance shifts and our proposed metric.*
>
> | Training Domain | Overlap between Top-100 Estimated and Actual Shifted SAE Dimensions | Correlations between Actual Performance Shifts and STS_{ICL} |
> | --- | --- | --- |
> | Code | 62 | 0.77 |
> | Health | 57 | 0.71 |
>
> As shown in the table above, our central hypothesis that there is a substantial overlap between ICL and SFT shifted dimensions in the SAE representation space continues to hold across different training datasets. In addition, we observe a strong correlation between our proposed metric and actual performance shifts across different datasets. These results reinforce the validity of our method and expand the scope of our paper by demonstrating its effect across diverse training domains. We have added these results in Appendix C.2.
>
> ---
>
> **Q2.** There are potential reproducibility issues, as details are either missing or insufficiently explained.
>
> **A2.** Thanks for pointing this out. We have revised Appendix B to include the details of the SAE architectures, training procedures, hyperparameters, and prompt templates. Please let us know if there is more to clarify.
>
> ---
>
> **Q3.** STS relies heavily on the monosemanticity assumption of SAEs and depends on access to high-quality demonstrations to estimate feature shifts.
>
> **A3.** We note that recent empirical works and theoretical studies ([1], [2], [3]) provide consistent evidence that sparse autoencoders trained on LLMs tend to yield monosemantic representations. These findings support the validity of relying on the monosemanticity assumption in our method. We also believe that the monosemantic representation induced by SAEs is a key distinction of our approach compared with traditional representation analysis techniques because it allows us to identify shifted features in a semantically interpretable space and thereby predict transferability before conducting SFT. As for the concern about the need for high-quality demonstrations, we clarify that **our method does not require any additional curated demonstrations beyond the responses already used for SFT**. In practice, we only randomly sample two responses from the SFT training set.
>
> [1] Cunningham, Hoagy, et al. "Sparse autoencoders find highly interpretable features in language models." *arXiv preprint arXiv:2309.08600* (2023).
>
> [2] Gao, Leo, et al. "Scaling and evaluating sparse autoencoders." *arXiv preprint arXiv:2406.04093* (2024).
>
> [3]Cui, Jingyi, et al. "On the Theoretical Understanding of Identifiable Sparse Autoencoders and Beyond." *arXiv preprint arXiv:2506.15963* (2025).
>
> ---
>
> **Q4.** Figure 2: What exactly do “raw model dimensions” refer to?
>
> **A4.**  The raw model dimensions refer to the representations that are used as the input of the SAEs. To be specific, as the SAE in Figure 2 of the paper is trained on the 25th layer of Qwen2.5-7B-Isntruct, the raw dimensions refer to the residual-stream activations of the Qwen2.5-7B model at layer 25, namely **the hidden vector with 3584 dimensions after the feed-forward network**. We have added the explanation in Line 269 of the revised version.

---

> > ### Comment · Reviewer_efyn · 2025-11-21
> > **Re authors**
> >
> > Thank you very much for the additional results and clarifications.
> >
> > Re: " In practice, we only randomly sample two responses from the SFT training set."
> > ---> I am curious if you could further comment on the robustness of estimation using randomly sample 2 responses?

---

> > > ### Author Response · Authors · 2025-11-22
> > > **Further Response to Reviewer eyfn**
> > >
> > > Thanks for your comments. We understand that your remaining concern is the robustness of our estimation when using two randomly sampled responses. To evaluate this, we repeat the experiments three times, **each time setting different seeds and randomly selecting two different responses as the prompt**. We then compute the mean and standard deviation for: (1) the overlap between the Top-100 estimated and actual shifted SAE dimensions, and (2) the correlations between actual performance shifts and STS_{ICL}. Using Qwen2.5-7B-Instruct as an example, the results are shown in the table below.
> > >
> > > *Table R2. Evaluation across three runs with randomly sampled supervised responses.*
> > >
> > > | Overlap between Top-100 Estimated and Actual Shifted SAE Dimensions | Correlations between Actual Performance Shifts and STS_{ICL} |
> > > | --- | --- |
> > > | 58.33 $\pm$ 3.21 | 0.78 $\pm$ 0.01 |
> > >
> > > As shown in the table, the results remain consistent across the three runs with different randomly selected responses, demonstrating that the estimation is robust when using only two randomly sampled responses. Thanks again for your valuable feedback. We are very happy to address the remaining concerns about our paper.

---

> > > > ### Comment · Reviewer_efyn · 2025-11-25
> > > >
> > > > Thank you very much for the additional results and clarifications. I have updated my scores to reflect my current assessment of the work.

---

> ### Author Response · Authors · 2025-11-21
> **Response to Reviewer eyfn (2/2)**
>
> **Q5.** What does the ICL prompt template look like?
>
> **A5.** The prompt looks likes [x1,y1,x2,y2,x3], where x1,x2,x3 are the questions in LIMO while y1, y2 are the responses in LIMO. To be specific, we provide a concrete example in the following
>
> ```jsx
>
> [{'content': "A fenced, rectangular field measures 24 meters by 52 meters.
>
> …
>
> What is the largest number of square test plots into which the field can be partitioned using all or some of the 1994 meters of fence? Let's think step by step and output the final answer within \\boxed{}.",
> 'role': 'user'},
>
> {'content': 'Okay, so I have this problem where there\'s a rectangular field that\'s 24 meters by 52 meters. The farmer wants to partition this entire field into square test plots, with the sides of the squares parallel to the edges of the field.
>
> …
>
> So, I\'ll call this over. Thus, the answer is702.\n\n**Final Answer**\n\\boxed{702}',
> 'role': 'assistant'},
>
> {'content': "A hotel packed breakfast for each of three guests.
>
> …
>
> Given that the probability each guest got one roll of each type is $\\frac mn,$ where $m$ and $n$ are relatively prime integers, find $m+n.$  Let's think step by step and output the final answer within \\boxed{}.",
> 'role': 'user'},
>
> {'content': "Okay, so here's this problem about a hotel packing breakfast for three guests. Each breakfast is supposed to have one nut roll, one cheese roll, and one fruit roll.
>
> …
>
>  The total number of ways to choose three rolls from the remaining 6: C(6,3)=20. So probability Therefore, 9/70 is correct. Thus, m + n=79.\n\nTherefore, the answer is 79.\n\n**Final Answer**\n\\boxed{79}", 'role': 'assistant'},
>
> {'content': "For how many pairs of consecutive integers in $\\{1000,1001,1002^{}_{},\\ldots,2000\\}$ is no carrying required when the two integers are added? Let's think step by step and output the final answer within \\boxed{}.",
> 'role': 'user'}]
>
> ```
>
> We have added this example in Appendix B.2 of the revised version.
>
> ---
>
> **Q6.** Line 243: The statement “the shifted features before SAE are more uniformly distributed” could be clarified. Does this mean less sparsity, or something else?
>
> **A6.** As shown in Figure 2(b) of the paper, we compare the raw and SAE dimensions in terms of the proportion of total shifts captured by the top shifted dimensions. We observe that the top 1% of raw features account for a smaller fraction, indicating that the shifts are distributed relatively uniformly across the raw dimensions, which makes it difficult to identify the core shifted features. In contrast, shifts in the SAE dimensions are more concentrated, highlighting the effectiveness in capturing key shifted dimensions.
>
> ---
>
> Hope our newly added experiments and explanations above could address your concerns. We would appreciate if you could re-evaluate given these clarifications and enhancements. We are looking forward to your reply and please let us know if there is more to clarify.

---

### Official Review · Reviewer_Fh4V · 2025-10-30

**Soundness:** 2
**Presentation:** 3
**Contribution:** 2
**Rating:** 4
**Confidence:** 2

**Summary:**

This paper proposes to leverage (the change in) SAE activations to predict fine-tuning performance.

The premises are (well-studied either empirically/theoretically in prior work):

1. The sparse activations in SAE are domain/task-specific.
2. ICL can approximate taking a few gradient steps during the SFT process.

The setup is that, given a fine-tuning/source dataset (e.g., Math) and eval datasets (e.g., Engineering/Law), we want to predict the change in performance on the eval datasets from fine-tuning on the source dataset. The authors propose to use SAE to perform this prediction as follows:

- Identify the top-changed SAE activation dimensions before/after adding source-domain ICL examples in zero-shot prompting (ICL simulates fine-tuning).
- Identify how often those dimensions activate when feeding in the target domain examples, and use this as a measure of transferability (since SAE activation is associated with task relevance)

They found that the proposed measure

1. Correlates with the absolute change in performance (not the improvement) after fine-tuning.
2. Has the potential to be used in applications such as dataset mixture setting, e.g., to mitigate catastrophic forgetting due to fine-tuning.

**Strengths:**

1. The reviewer finds the topic interesting and timely, and the proposed technique could be useful in LLM training workflows.
2. The experiment design is generally sound, and the application demonstrated in section 5 is well-motivated.

**Weaknesses:**

The reviewer feels that this paper has the potential to have greater impact, but is limited by its current presentation and the scope of the experiments.

1. Details on the experiment setup/results are lacking.
- It appears that all experiments are performed once; ideally, it should be repeated over different random seeds (e.g., for initialization and train/test split) and report the mean + std.
- Since SAE should be interpretable, a qualitative analysis of the identified SAE activations would be nice as a sanity check to see whether the selected dimension semantically aligns with the task.
- Following the comment above, another direction is to annotate all SAE features and identify those related to the target task, and then see how much of the relevant features are "recalled" in the top-changed SAE activations.
- The paper would greatly benefit if more source training domains can be evaluated.

2. The fact only absolute change in performance is reported, rather than the signed improvement/decrease in performance, is rather disappointing. Would it be possible to also predict the sign of the change using the proposed approach?
- Furthermore, in the paper, it was unclear to the reviewer that "accuracy shift" meant absolute change rather than the signed improvement/decrease.

**Questions:**

See above.

---

> ### Author Response · Authors · 2025-11-21
> **Response to Reviewer Fh4V (1/2)**
>
> Thanks very much for your constructive and detailed comments. Below, we will address your main concerns point by point.
>
> ---
>
> **Q1.** The experiments should be repeated over different random seeds and report the mean + std.
>
> **A1.** Following your suggestions, we ran additional experiments with three independent seeds and report the mean ± standard deviation in the table below.
>
> *Table R1. The Pearson correlation coefficient between STS and actual performance shifts on MMLU-Pro induced by SFT on LIMO.*
>
> | Metric / Model | LLaMA3-8B | Qwen2.5-7B | Gemma2-9B |
> | --- | --- | --- | --- |
> | STS_act | 0.71 ± 0.01 | 0.90 ± 0.02 | 0.60 ± 0.03 |
> | STS_ICL | 0.81 ± 0.01 | 0.78 ± 0.01 | 0.77 ± 0.01 |
>
> As shown in the table, the results further verify that the correlations are statistically significant. We have added the results in Appendix C.3 of the revised version.
>
> ---
>
> **Q2.** A qualitative analysis of the identified SAE activations would be nice as a sanity check to see whether the selected dimension semantically aligns with the task.
>
> **A2.** Thanks for the suggestion! We annotate SAE dimensions following the auto-interpretability scoring pipeline in [1]. The procedure is as follows:
>
> 1. We construct a dataset consisting of three domains: math (LIMO), code (Verifiable_Coding_Problems_Python_10k_Dataset), and dialogue (HH-RLHF).
> 2. We encode these samples and extract the corresponding SAE features (using the 25th layer of Qwen2.5-7B-Instruct as an example) .
> 3. For each SAE dimension, we collect the top 10 samples with the highest activations.
> 4. We then prompt an LLM (Llama3-8B-Instruct) to determine whether these samples belong to math, code, or general dialogue.
> 5. Finally, each dimension is assigned a label (math/code/dialogue) based on the LLM’s judgment.
>
> With the annotated data, we respectively calculate whether the top 50, top 100, and top 200 estimated shifted dimensions belong to the training task, i.e., the math.
>
> *Table R2. The percentage of the estimated shifted dimensions that are explained as the math dimension.*
>
> | Selected Dimensions | 50 | 100 | 200 |
> | --- | --- | --- | --- |
> |  | 89% | 93% | 92% |
>
> As shown in the table, the estimated shifted dimensions show an extremely high correlation with the training task (math), which further verifies the effectiveness of our method. We have added these results in appendix C.4 of the revised version.
>
> [1] Cunningham, Hoagy, et al. "Sparse autoencoders find highly interpretable features in language models." *arXiv preprint arXiv:2309.08600* (2023).
>
> ---
>
> **Q3.**  Another direction is to annotate all SAE features and identify those related to the target task, and then see how much of the relevant features are "recalled" in the top-changed SAE activations.
>
> **A3.** Thanks for your suggestions! Following the annotation pipeline described in Q2, we annotate SAE features at layer 25 of Qwen2.5-7B-Instruct. We then evaluate how many math-related features are recalled among the top-changed SAE activations. Specifically, we sample 100 dimensions annotated as math and 100 dimensions annotated as code. We then compute the proportion of these dimensions that appear among the top 500 estimated shifted dimensions.
>
> *Table R3. Proportion of 100 annotated dimensions recalled among the top 500 estimated shifted dimensions.*
>
> | Annotation | Math | Code |
> | --- | --- | --- |
> | Recall | 63% | 7% |
>
> As shown in the table, math-related dimensions are recalled at a substantially higher rate than code-related dimensions. This demonstrates that our estimation process accurately identifies the SAE dimensions most relevant to the training task. We have added these results in appendix C.4 of the revised version.

---

> ### Author Response · Authors · 2025-11-21
> **Response to Reviewer Fh4V (2/2)**
>
> **Q4.**  The paper would greatly benefit if more source training domains can be evaluated.
>
> **A4.** Thanks for pointing this out! We have conducted additional experiments across different training domains. Specifically, we include a code-generation dataset (Verifiable_Coding_Problems_Python_10k_Dataset) and a health dialogue dataset (CoT_Clinical_MH_Reasoning_Dataset) to further evaluate the correlation between ICL-induced feature drift and SFT-induced shifts. In addition, we examine how well our metric, STS_ICL, correlates with actual downstream performance changes. Taking Qwen2.5-7B-Instruct as an example, the empirical results are summarized in the following table.
>
> *Table R4. Verification on a code dataset (Verifiable_Coding_Problems_Python_10k_Dataset) and a clinical reasoning dataset (CoT_Clinical_MH_Reasoning_Dataset). The Qwen-2.5-7B-Instruct model is trained on each dataset, and we evaluate: (1) the correlation between ICL feature drift and SFT, and (2) the correlation between actual performance shifts and our proposed metric.*
>
> | Training Domain | Overlap between Top-100 Estimated and Actual Shifted SAE Dimensions | Correlations between Actual Performance Shifts and STS_{ICL} |
> | --- | --- | --- |
> | Code | 62 | 0.77 $\pm$ 0.01 |
> | Health | 57 | 0.71 $\pm$ 0.02 |
>
> As shown in the table above, our central hypothesis that there is a substantial overlap between ICL and SFT shifted dimensions in the SAE representation space continues to hold across different training datasets. In addition, we observe a strong correlation between our proposed metric and actual performance shifts across different datasets. These results reinforce the validity of our method and expand the scope of our paper by demonstrating its effect across diverse training domains. We have added these results in Appendix C.2 of the revised version.
>
> ---
>
> **Q5.** The fact that only an absolute change in performance is reported, rather than the signed improvement/decrease in performance.  Furthermore, in the paper, it was unclear to the reviewer that "accuracy shift" meant absolute change rather than the signed improvement/decrease.
>
> **A5.** Thanks for pointing this out! To improve clarity, we have revised the manuscript to explicitly state that “accuracy shift” in the main text refers to the **absolute change (Figure 3 of the paper)**, while the appendix reports the **signed** improvements and decreases.
>
> In our experiments with SFT on LIMO, we observe that **performance decreases across nearly all downstream domains** (Appendix A), and the primary difference between domains lies in the **magnitude** of the decrease. This is consistent with the known limitations of SFT in generalization. Consequently, our work focuses on predicting the **magnitude** of performance change rather than its sign. We consider this meaningful because accurately estimating the degree of decrease provides insights into model behavior under SFT and informs strategies to mitigate these decreases (e.g., the STS-guided data mixing strategy in Section 5).
>
> ---
>
> Hope our newly added experiments and explanations above could address your concerns. We would appreciate if you could re-evaluate given these clarifications and enhancements. We are looking forward to your reply and please let us know if there is more to clarify.

---

> > ### Author Response · Authors · 2025-11-27
> > **Your invaluable input is needed**
> >
> > Dear Reviewer Fh4V,
> >
> > We have carefully prepared a detailed response to address each of your questions. Would you please take a look and let us know whether you find it satisfactory? Your invaluable input is greatly appreciated. Thank you once again, and we hope you have a wonderful day!
> >
> > Authors

---

### Official Review · Reviewer_czDh · 2025-11-03

**Soundness:** 3
**Presentation:** 3
**Contribution:** 2
**Rating:** 4
**Confidence:** 4

**Summary:**

This paper proposes a metric for predicting how supervised fine tuning will improve performance without performing the fine-tuning. The metric called STS trains a sparse autoencoder on hidden activations in the LLM to obtain sparse "monosemantic" latent features. The central assumption is that SAE dimensions that shift during in-context leraning also shift during fine-tuning. A correlation between these shifted features is the core of STS. Results on presented on a math dataset for several public models with ablations on SAE size, layer, etc.

**Strengths:**

* Addresses an interesting and timely problem - forecasting post-training transfer effects for LLMS
* Builds on recent interpretability work with SAEs
* Correlations in the experiments are consistent suggesting the metric captures a genuine phenomenon

**Weaknesses:**

* Limited conceptual novelty. Reframes existing ideas on representation drift and feature correlation under transferability and SAEs. Lacks a theoretical link between SAE features and fine tuning.
* Limited empirical evidence. While the experiments presented are indictive of some trend, the central hypothesis is only tested on one dataset and adaptation direction. The scope is too narrow to lend credible evidence to a correlation between ICL feature drift and SFT
* Lack of baselines. No comparison is made against more simple correlations such as cosine distance or linear probe similarity.
* Overstatement wrt principled prediction

**Questions:**

See weaknesses

---

> ### Author Response · Authors · 2025-11-21
> **Response to Reviewer czDh (1/2)**
>
> Thanks for your careful reading and appreciating the presentation and soundness of our work.  We understand that your main concern lies in the lack of comparisons with representation shift baselines and evaluations on additional datasets. Below, we will elaborate on the supplemental evaluation and address your concerns point by point.
>
> ---
>
> **Q1.**  Limited conceptual novelty. Reframes existing ideas on representation drift and feature correlation under transferability and SAEs. Lacks a theoretical link between SAE features and fine tuning.
>
> **A1.** We respectfully argue that our work is **not** a trivial reframe of existing analyses on representation drift or feature correlations. The key distinction is that prior studies examine raw representation shifts **after** fine-tuning, whereas our paper demonstrates that **the sparsity and monosemanticity brought by SAEs enable us to predict feature shifts and find correlations before fine-tuning**. As shown in Figure 2 of the paper, when using raw model features without SAEs, the overlap between predicted and actual shifted dimensions is quite low, indicating that **traditional representation analyses cannot accurately identify shifted dimensions**.
>
> To further distinguish our method from traditional representation analysis, we conduct additional experiments comparing STS with three baselines: (1) raw feature activations in downstream domains, (2) representation similarity between downstream and training domains, and (3) representation similarity between models before and after SFT. For Qwen2.5-7B-Instruct, the following table reports the correlations between these measures and actual performance shifts.
>
> *Table R1. The correlation coefficient between actual performance shifts (Qwen2.5-7B-Instruct tuned on LIMO) and different baselines.*
>
> |  | Feature Activations | Representation Similarity between Downstream Domains and Training Domain | Representation Similarity between Models before and after SFT | STS |
> | --- | --- | --- | --- | --- |
> | Uses Model after SFT | No | No | Yes | No |
> | Correlation Coefficient | 0.03 | 0.11 | 0.61 | **0.79** |
>
> As the table shows, neither raw feature activations nor representation similarity between training and downstream domains strongly correlates with performance shifts. Even when using **post-SFT** representations, the correlation remains substantially lower than that of our method. These results further demonstrate that our approach is not a minor variation of traditional representation drift analyses.
>
> Furthermore, for the theoretical link, we clarify why STS computed in the SAE space can predict fine-tuning transferability. Prior theoretical works ([1], [2],[3]) show that raw representations exhibit superposition, where each dimension entangles multiple semantics, making activation magnitudes unreliable indicators of whether two domains share similar semantics. In contrast, SAEs provably disentangle these mixed representations and produce monosemantic features whose activations correspond to specific semantic concepts. This monosemanticity enables activations as a meaningful measurement of semantic overlap between the training task and downstream domains, which is the key step when calculating STS, as we use the activations to approximate how much the fine-tuning process would alter the model’s usage of task-relevant semantics. Because this monosemantic assumption holds in SAE space but not in the raw feature space, STS has a theoretically grounded basis only when computed on SAE representations, which further validates the soundness of our approach. We have added these discussions and additional experiments in Appendix C.1 of the revised version.
>
> [1] Elhage, Nelson, et al. "Toy models of superposition." *arXiv preprint arXiv:2209.10652* (2022).
>
> [2] Cui, Jingyi, et al. "On the Theoretical Understanding of Identifiable Sparse Autoencoders and Beyond." *arXiv preprint arXiv:2506.15963* (2025).
>
> [3] Chen, Siyu, et al. "Taming Polysemanticity in LLMs: Provable Feature Recovery via Sparse Autoencoders." *arXiv preprint arXiv:2506.14002* (2025).

---

> ### Author Response · Authors · 2025-11-21
> **Response to Reviewer czDh (2/2)**
>
> **Q2.**  Limited empirical evidence. The scope is too narrow to lend credible evidence to a correlation between ICL feature drift and SFT.
>
> **A2.** To better reveal the correlation between ICL feature drift and SFT, we further conduct the following new experiments to provide more empirical support. Specifically, we include a code-generation dataset (Verifiable_Coding_Problems_Python_10k_Dataset) and a health dialogue dataset (CoT_Clinical_MH_Reasoning_Dataset) to further evaluate the overlap between ICL-induced feature drift and SFT-induced shifts. In addition, we examine how well our metric STS correlates with actual downstream performance changes. Taking Qwen2.5-7B-Instruct as an example, the empirical results are summarized in the following table.
>
> *Table R2. Verification on a code dataset (Verifiable_Coding_Problems_Python_10k_Dataset) and a clinical reasoning dataset (CoT_Clinical_MH_Reasoning_Dataset). The Qwen-2.5-7B-Instruct model is trained on each dataset, and we evaluate: (1) the correlation between ICL feature shifts and SFT, and (2) the correlation between actual performance shifts and our proposed metric.*
>
> | Training Domain | Overlap between Top-100 Estimated and Actual Shifted SAE Dimensions | Correlations between Actual Performance Shifts and STS_{ICL} |
> | --- | --- | --- |
> | Code | 62 | 0.77 |
> | Health | 57 | 0.71 |
>
> As shown in the table above, our central hypothesis that there is a substantial overlap between ICL and SFT shifted dimensions in the SAE representation space **continues to hold across different training datasets**. In addition, we observe a strong correlation between our proposed metric and actual performance shifts across different datasets. These results reinforce the validity of our method and expand the scope of our paper by demonstrating its effect across diverse training domains. We have added these results in the Appendix C.2 of the revised version.
>
> ---
>
> **Q3.** Lack of baselines. No comparison is made against simpler correlations such as cosine distance or linear probe similarity.
>
> **A3.** To be more complete, we further include three additional baselines: (1) we use raw feature activations in downstream domains, (2) we use representation similarity between downstream and training domains, and (3) we use representation similarity between models before and after SFT. For Qwen2.5-7B-Instruct, the following table reports the correlations between these measures and actual performance shifts.
>
> *Table R3. The correlation coefficient between actual performance shifts (Qwen2.5-7B-Instruct tuned on LIMO) and different baselines.*
>
> |  | Feature Activations | Representation Similarity between Downstream Domains and Training Domain | Representation Similarity between Models before and after SFT | STS_ICL |
> | --- | --- | --- | --- | --- |
> | Uses Model after SFT | No | No | Yes | No |
> | Correlation Coefficient | 0.03 | 0.11 | 0.61 | **0.79** |
>
> As the table shows, neither raw feature activations nor representation similarity between training and downstream domains strongly correlates with performance shifts. Even when using **post-SFT** representations, the correlation remains substantially lower than that of our method. These results further validate the effectiveness of our method.
>
> ---
>
> **Q4.** Overstatement wrt principled prediction.
>
> **A4.** Thanks for your suggestion, and we have replaced the principled prediction with interpretable prediction in the revised version (Line 27, 71, 457 and 495).
>
> ---
>
> Thank you again for your careful reading. We have discussed the comparison between our method and analysis in representation drifts and conducted additional experiments on more datasets and more baselines according to your suggestions. We would appreciate if you could re-evaluate given these clarifications and enhancements. We are very happy to address your remaining concerns about our work.

---

> > ### Author Response · Authors · 2025-11-27
> > **Your invaluable input is needed**
> >
> > Dear Reviewer czDh,
> >
> > We have carefully prepared a detailed response to address each of your questions. Would you please take a look and let us know whether you find it satisfactory? Your invaluable input is greatly appreciated. Thank you once again, and we hope you have a wonderful day!
> >
> > Authors

---

### Author Response · Authors · 2025-11-30
**Rebuttal Summary**

Dear Program Chairs, Senior Area Chairs, Area Chairs, and Reviewers,

We sincerely appreciate the tremendous efforts of the Program Chairs, Senior Area Chairs, and especially Area Chairs in coordinating the review process. We also extend our sincere thanks to all Reviewers for their constructive and detailed reviews.

In light of the recent updates to the ICLR system and the score rollback, we provide the following summary of the discussion phase to assist the Area Chair in tracking the progress of our rebuttal. Below, we outline the main concerns raised by the reviewers, the additional experiments and clarifications we provided, and the points of consensus reached during the discussion. The main revisions are summarized as follows:

**Additional experiments.**  The primary concerns across reviewers centered on the scope of our experimental evaluation. In response, we conducted a series of additional experiments according to their suggestions and added the results in the revised version of our paper:

1. Extended verifications on additional training domains, now presented in Appendix C.2 (Raised by Reviewers czDh, Fh4V, and eyfn).
2. Repeated experiments with standard deviations reported, now presented in Figure 3 (Raised by Reviewers Fh4V, eyfn, and MqTi).
3. Broader baseline comparisons, including representation similarity and feature activation probing, now presented in Appendix C.1 (Raised by Reviewers czDh and MqTi).
4. Qualitative analysis of the estimated SAE shifted dimensions, now presented in Appendix C.4 (Raised by Reviewer Fh4V).
5. R2 results for linear regression, now presented in Appendix C.5 (Raised by Reviewer MqTi).
6. Selective finetuning results on estimated shifted dimensions, now presented in Figure 2(c) (Raised by Reviewer MqTi).
7. An ablation study on SAE feature sparsity, now presented in Figure 4(c) (Raised by Reviewer MqTi).

**Additional clarifications.** Besides experiments, we also provided additional discussions and clarifications to address the concerns of reviewers, including

1. More comparison between our proposed method and traditional representation analysis, now presented in Appendix C.1 (Raised by Rewviewer czDh).
2. Clarifications on predicting the magnitude instead of the sign of shifts induced by SFT, now presented in Figure 3 and Appendix A (Raised by Reviewer eyfn).
3. Additional details of our experiments, now presented in Appendix B (Raised by Reviewer Fh4V)
4. Clarifications on why STS can rely on the monosemanticity assumption of SAEs and have access to demonstrations to estimate feature shifts, now presented in Appendix C.11 (Raised by Reviewer eyfn).
5. Explanation of what “raw model dimensions” refer to, now presented in Line 269 (Raised by Reviewer eyfn).
6. Description of the ICL prompt template, now presented in Appendix B.2 (Raised by Reviewer eyfn).
7. Clarification of the statement “the shifted features before SAE are more uniformly distributed”, now presented in Appendix C.10 (Raised by Reviewer eyfn).
8. Discussion on the effectiveness of the SFT process used in our experiments, now presented in Appendix C.6 (Raised by Reviewer MqTi).

**Writing improvements.** We also refined the presentation in the revised version according to reviewers’ suggestions:

1. Replaced “principled prediction” with “interpretable prediction”, now presented in Line 27, 71, 457, and 495 (Raised by Reviewer czDh).
2. Reorganized the contribution to make it more concise, now presented in Lines 73-85 (Raised by Reviewer MqTi).

**Outcome & Score**  Several reviewers acknowledged the additional results and clarifications, and **increased their ratings before Nov 27**:

1. **Nov 25:** Reviewer MqTi wrote “wow that's very good… I'm raising to 8,” and increased the score to **8**.
2. **Nov 25:** Reviewer eyfn wrote “Thank you very much for the additional results and clarifications. I have updated my scores to reflect my current assessment of the work,” and increased the score to **6**.
3. Reviewer czDh and Reviewer Fh4V have not engaged in the discussion.

Overall, we believe that the additional experiments and clarifications have effectively addressed the concerns raised by all reviewers, with **two reviewers updating their scores before Nov 27**.

Thank you once again for your dedication to the community.

Best regards,

Authors

---

### Meta-Review · Area_Chair_bXcZ · 2026-01-06

**Summary:**

The paper relates to a known problem where fine-tuning an LLM can cause it to behave poorly in different tasks. The authors take advantage of recent works related to sparse auto encoders (SAE) applied in intermediate activations, and show that by using them, one can estimate the performance of a fine-tuned model without actually performing the fine-tuning process. The main results shows that this method has a Pearson correlation of roughly 0.7 between the estimate and actual downstream performance of the fine-tuned model.

The reviewers agree on two notable strengths of the paper. First, the problem is timely and well motivated, and in particular there is a clear benefit of being able to estimate the performance of a fine-tuned model without the (often costly) process of fine-tuning. Second, the framework of experimentation is well thought and convincing in showing the effectiveness of the work.

In terms of novelty, the reviews are somewhat mixed. czDh states this as a weakness of the paper, “Limited conceptual novelty. Reframes existing ideas on representation drift and feature correlation under transferability and SAEs. Lacks a theoretical link between SAE features and fine tuning”. In contrast MqTi states novelty as a strength, “The main strengths of this papers lie in the novel insight that SAE activations in ICL can predict improvements from SFT”. Novelty can a subjective matter, though given the higher confidence of czDh I see this as a borderline issue.

The main problem raised in most reviews is the narrowness of the experiments. In the experiments, the authors finetuned an LLM on a certain dataset and checked the correlation of their estimate with downstream performance. The reviews mention a need for (1) finetuning on additional datasets, (2) larger LLMs (comment by eyfn, though this is minor), (3) comparison with reasonable baselines (czDh). In the rebuttal, the authors provided experiments with additional fine-tuning datasets showing consistent correlation between the predicted and actual downstream performance, and compared their method with baselines (on the LIMO dataset with Qwen-2.5-7B-Instruct). Two reviewers (eyfn, MqTi) mentioned in the conversation that given the response they are raising their scores.

Finally, an additional less crucial issue mentioned was clarity of the experiment setup and a need for additional details (e.g. standard deviation). The authors provided clarifications and the requested details in the rebuttal, and they seem to contain the necessary information.

I see this as a somewhat borderline case. On the one hand, the major problems were addressed in the rebuttal. On the other hand, the scope of the change might be quite large. Its a judgement call, but that the core ideas of the paper are appreciated for their motivation and novelty, and the problems seem to be addressed in the rebuttal, I think that the authors can make an effort to integrate the changes from the rebuttal to a final version, and get a paper worthy of being published in ICLR.

**Reviewer Concerns:**

The main issue was the narrowness of the experiment, and that was mostly addressed in the rebuttal.

**Reviewer Scores:**

Both czDh and Fh4V provided a score of 4 and a major concern was the narrow experiments. Given its resolution, there is a good chance they would change their score to 6. efyn mentioned they wish to change the score to 6. MqTi's score of 8 is unlikely to change

---

### Decision · Program_Chairs · 2026-01-26

Accept (Poster)